# A human progeria-associated BAF-1 mutation modulates gene expression and accelerates aging in *C. elegans*

Raquel Romero-Bueno [ID]1, Adrián Fragoso-Luna [ID]1, Cristina Ayuso [ID]1, Nina Mellmann [ID]1, Alan Kavsek [ID]2, Christian G Riedel [ID]2, Jordan D Ward [ID]3 & Peter Askjaer [ID]1✉

## Abstract

**Alterations in the nuclear envelope are linked to a variety of rare diseases termed laminopathies. A single amino acid substitution at position 12 (A12T) of the human nuclear envelope protein BAF (Barrier to Autointegration Factor) causes Néstor-Guillermo Progeria Syndrome (NGPS). This premature ageing condition leads to growth retardation and severe skeletal defects, but the underlying mechanisms are unknown. Here, we have generated a novel in vivo model for NGPS by modifying the *baf-1* locus in *C. elegans* to mimic the human NGPS mutation. These *baf-1(G12T)* mutant worms displayed multiple phenotypes related to fertility, lifespan, and stress resistance. Importantly, nuclear morphology deteriorated faster during aging in *baf-1(G12T)* compared to wild-type animals, recapitulating an important hallmark of cells from progeria patients. Although localization of BAF-1(G12T) was similar to wild-type BAF-1, lamin accumulation at the nuclear envelope was reduced in mutant worms. Tissue-specific chromatin binding and transcriptome analyses showed reduced BAF-1 association in most genes deregulated by the *baf-1(G12T)* mutation, suggesting that altered BAF chromatin association induces NGPS phenotypes via altered gene expression.**

**Keywords** Aging; *BANF1*; NGPS; Nuclear Lamina; Stress Resistance
**Subject Categories** Chromatin, Transcription & Genomics; Genetics, Gene Therapy & Genetic Disease

## Introduction

The chromosomes in eukaryotic cells are physically enclosed by the nuclear envelope (NE), which protects them from mechanical stress and serves as an anchoring point for chromatin and associated factors (Manzo et al, 2022). The NE is composed of outer and inner nuclear membranes, the nuclear lamina, and nuclear pore complexes to facilitate nucleocytoplasmic transport. The nuclear lamina is a rigid meshwork that consists mainly of intermediate filament proteins called lamins as well as a variety of lamin-associated proteins, including integral inner nuclear membrane proteins. Nuclear organization is dynamic and changes during development and aging. Moreover, chromatin is differently organized in the diverse cell types within an organism (Ungricht and Kutay, 2017). Furthermore, both environmental stress factors (such as temperature, pathogens, or nutrient availability) and internal factors (like cellular stress and mutations) can provoke changes in nuclear organization and gene expression. It is therefore not surprising that NE alterations are related to dramatic human conditions and attract intensive biomedical research efforts (Dobrzynska et al, 2016b; Shin and Worman 2022; Fragoso-Luna and Askjaer, 2023).

Barrier to Autointegration Factor 1 (BAF/BANF1) is a small (89 amino acids) chromatin-binding protein enriched at the NE and highly conserved across metazoans (Sears and Roux, 2020). BAF acts as a homodimer that binds simultaneously to DNA, lamins, and inner nuclear membrane LEM (LAP2, emerin, MAN1) domain proteins, thus serving as an anchoring factor (Bradley et al, 2005; Samson et al, 2018). BAF was initially discovered as a factor required for retroviral infectivity (Lee and Craigie, 1998). Its depletion is lethal during embryogenesis in *Caenorhabditis elegans* and *Drosophila melanogaster* (Zheng et al, 2000; Furukawa et al, 2003; Margalit et al, 2005) and decreases the survival of mouse and human embryonic stem cells (Cox et al, 2011). The localization of BAF at the NE is interdependent of its interaction partners, including lamins (Margalit et al, 2005; Lin et al, 2020), and LEM domain proteins, although BAF is also present in the nucleoplasm and the cytoplasm (Haraguchi et al, 2001; Margalit et al, 2007). BAF localization during the cell cycle is controlled by phosphorylation: at the entry to mitosis vaccinia-related kinase 1 (VRK1) phospholylates BAF, which reduces its affinity for chromatin (Nichols et al, 2006; Marcelot et al, 2021). The protein phosphatase 2 (PP2A) complex dephosphorylates BAF at the end of mitosis, allowing reassociation of BAF to chromatin and NE components (Asencio et al, 2012). Interfering with BAF phosphorylation has severe consequences: when VRK1 is depleted from *C. elegans*

¹Andalusian Centre for Developmental Biology, Consejo Superior de Investigaciones Científicas (CSIC), Universidad Pablo de Olavide, Junta de Andalucía, Carretera de Utrera, km 1, 41013 Sevilla, Spain. ²Department of Biosciences and Nutrition, Karolinska Institutet, Huddinge 14157, Sweden. ³Department of Molecular, Cell, and Developmental Biology, University of California-Santa Cruz, Santa Cruz, CA 95064, USA. ✉E-mail: pask@upo.es

embryos or human cells, BAF remains associated with chromosomes when cells enter mitosis, which leads to defects in chromosome segregation (Gorjanacz et al, 2007; Molitor and Traktman, 2014). BAF is required for mitotic chromosome coherence and nuclear assembly toward the end of mitosis (Gorjanacz et al, 2007; Samwer et al, 2017). It localizes on the surface of the condensed chromosomes to maintain these tightly together, thereby ensuring the reassembly of a NE that encompasses all chromosomes in a single nucleus (Haraguchi et al, 2008; Samwer et al, 2017). More recently, BAF was implicated in the repair of NE ruptures (Halfmann et al, 2019). BAF localizes to the rupture site by binding to the leaking chromatin, and recruits LEM domain proteins to repair the NE and restore the permeability barrier (Young et al, 2020; Kono et al, 2022; Barger et al, 2023). Finally, BAF has also been involved in sensing mechanical stimuli to regulate cell cycle progression and DNA replication (Wang et al, 2018; Unnikannan et al, 2020).

About a decade ago, an identical homozygous amino acid substitution in human BAF (A12T) was identified in 2 patients (Néstor and Guillermo) suffering from premature aging symptoms, leading to the term Néstor-Guillermo progeria syndrome (NGPS) (Puente et al, 2011). NGPS shares phenotypic similarities with Hutchinson-Gilford progeria syndrome (HGPS), which has been described in much greater detail and is most often caused by mutation in *LMNA* that encodes lamin A/C (Gonzalo et al, 2017). However, cardiovascular deficiencies and metabolic complications have not been described for NGPS patients and the manifestation of the condition occurs later, approximately 24 months after birth (Cabanillas et al, 2011). Similarly to HGPS, nuclear morphology is abnormal in NGPS fibroblast, and can be restored by expression of wild-type BAF (Puente et al, 2011; Janssen et al, 2022). It was proposed that the mutation destabilizes BAF (Puente et al, 2011), although this has been questioned by others (Paquet et al, 2014). The residue mutated in NGPS is situated close to the interface with lamin and the mutation was reported to reduce the affinity between the two proteins (Samson et al, 2018; Janssen et al, 2022). However, another study found that ectopically expressed BAF A12T localized correctly to the NE, and that the mutation modulated the affinity of BAF for DNA, thus possibly interfering with chromatin anchoring at the NE (Paquet et al, 2014). The A12T mutation was demonstrated to also affect NE rupture repair. Although mutant BAF A12T accumulated at rupture sites allowing repair, nuclei were observed to suffer repeated rupture more frequently than in control cells (Janssen et al, 2022). Despite the identification of several cellular defects in NGPS cells, it remains unclear what eventually leads to the patient phenotypes after seemingly normal fetal and postnatal development.

Almost all studies described above were performed with cultured cells. Although the use of patient-derived cell lines is essential to answer many of the open questions about NGPS, cell lines suffer limitations in terms of interplay between tissues and different organs (Yamamoto et al, 2024). To study the effect of the NGPS mutation in a whole-animal model, we introduced the equivalent mutation in the endogenous *baf-1* locus in *C. elegans*. We show that *baf-1(G12T)* mutants recapitulate several phenotypes typical of progerias, including shortened lifespan and accelerated deterioration of nuclear morphology. Moreover, the mutants have reduced fertility, increased sensitivity to irradiation and temperature stress but, surprisingly, are more tolerant to oxidative stress.

We observed numerous changes in BAF-1's association with chromatin in hypodermal and intestinal cells, suggesting that the G12T mutation affects the affinity of BAF-1 for chromatin. By focusing on BAF-1's role in chromatin organization, we identified genes that are both deregulated and differentially bound by BAF-1(G12T). Enriched gene classes include structural components of cuticle and ribosomes, suggesting specific functions of BAF in the regulation of gene expression. We conclude that *baf-1(G12T)* mutants serve as a relevant model to advance our understanding of BAF function and the mechanisms underlying NGPS.

## Results

### The *baf-1(G12T)* mutation reduces brood size and lifespan

*C. elegans* and human BAF proteins share 74% amino acid sequence similarity (Margalit et al, 2005) and are predicted to adopt identical three-dimensional structures (Fig. EV1A), making the worm a suitable model to study BAF mutations. To evaluate the effect of the NGPS mutation on *C. elegans* health span parameters, we modified the endogenous *baf-1* locus to encode a threonine residue at position 12 instead of the naturally occurring glycine.

We first measured fertility. At 20 °C, *baf-1(G12T)* self-fertilizing hermaphrodites produced the same number of descendants as control worms (Fig. 1A). In contrast, when worms were shifted to 25 °C before reaching adulthood, their fertility was reduced by 66% (Figs. 1B and EV1B), whereas embryonic viability was unaffected (Fig. EV1C). We noticed that *baf-1(G12T)* hermaphrodites deposited many unfertilized oocytes (UFOs) on the plates at 25 °C (Fig. 1B). *C. elegans* hermaphrodites usually lay a few UFOs when they run out of sperm (produced during the larval L4 stage) at the end of their reproductive cycle (Collins et al, 2008). However, *baf-1(G12T)* mutants laid UFOs throughout the fertile period, representing a 34% of the total lay versus only 6% in control animals (Fig. 1B). We investigated if the UFOs laid by *baf-1(G12T)* hermaphrodites at 25 °C could be explained by a lack in sperm production. However, we found sperm with the same frequency in the spermatheca of control and *baf-1(G12T)* hermaphrodites on days 1 and 2 of adulthood (Fig. 1C,D). On day 3, sperm were still observed in approximately half of the control animals but in none of the *baf-1(G12T)* mutants. These data indicate that although sperm were produced there might be defects in function. To test this hypothesis, we mated *baf-1(G12T)* day 1 adult hermaphrodites with wild-type males at 25 °C for 7 h. After mating we did not observe UFOs and the brood size was comparable to self-fertilizing wild-type hermaphrodites (Fig. EV2A). Mating *baf-1(G12T)* hermaphrodites with *baf-1(G12T)* males also eliminated UFOs, but did not fully restore the brood size, suggesting a sperm defect in *baf-1(G12T)* males (Fig. EV2A). To test this possibility, we incubated *baf-1(G12T)* males with *fog-2(q71)* feminized worms that only produce oocytes and counted daily offspring. At 25 °C, the *fog-2(q71)* allele prevents spermatogenesis specifically in XX hermaphrodites whereas X0 males are unaffected (Schedl and Kimble, 1988). We observed a reduction in brood size of approximately one-third when sperm came from *baf-1(G12T)* males (Fig. EV2B,C). Thus, we concluded that the *baf-1(G12T)* mutation has a negative impact on spermatogenesis. The male/female ratio in the progeny was ~1, suggesting that meiotic segregation of chromosomes was normal in *baf-1(G12T)* males.

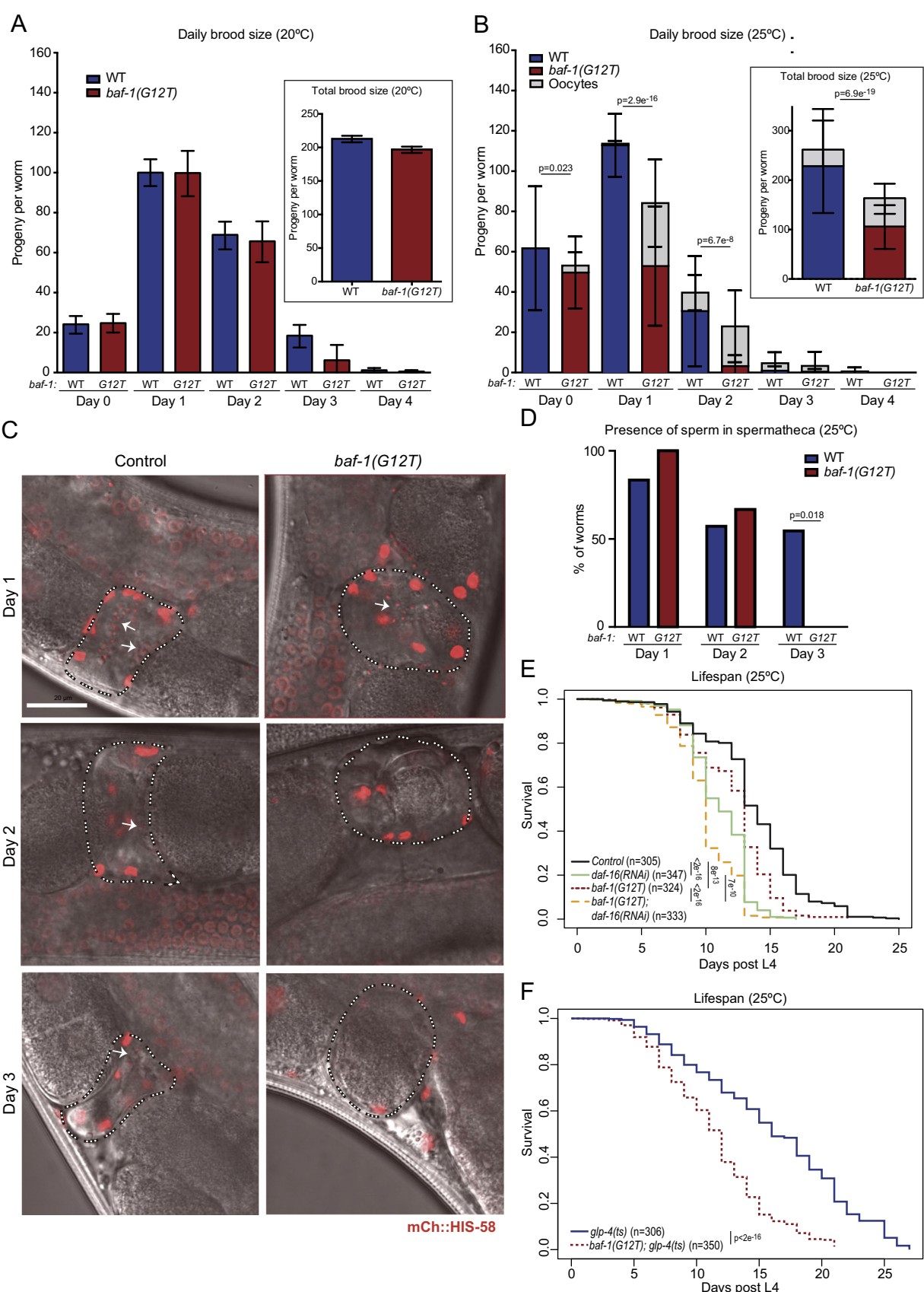

**Figure 1.  Temperature sensitive brood size and lifespan reduction caused by *baf-1(G12T)* mutation.**

(A, B) Daily and total brood size was evaluated at 20 °C (A) and at 25 °C (B). For the wild-type (WT; blue columns) control, 14 and 41 adults were analyzed at 20 °C and 25 °C, respectively, whereas for *baf-1(G12T)* (red columns) the equivalent numbers were 11 and 42. Day 0 corresponds to the 24 h interval from L4 to young adult and henceforth. Unfertilized oocytes (gray) were observed on *baf-1(G12T)* plates at 25 °C already on day 0 and on control plates from day 2. Columns and error bars represent mean and standard deviation, respectively. (C, D) Presence of sperm in the spermatheca of wild-type and *baf-1(G12T)* hermaphrodites grown at 25 °C and expressing mCh::HIS-58 was evaluated on days 1, 2, and 3. Representative images (C) with indication of spermatheca and sperm by dashed lines and arrows, respectively, and combined results (D) from at least 7 animals at each time point and each strain. Scale bar represents 20 µm. (E, F) Lifespan assay of wild-type and *baf-1(G12T)* hermaphrodites at 25 °C fed with either control or *daf-16* RNAi bacteria (E) or germline-ablated *glp-4(ts)* mutants combined with either wild-type *baf-1* or *baf-1(G12T)* (F). Data combined from 3 (E) or 4 (F) independent experiments; the number of animals is indicated in brackets. *P* values from two-sided t-tests (A, B) and Chi-square (D–F) tests are indicated; *p* values ≥ 0.05 are considered not significant and are not shown. *P* values in (B) refer to comparison of number of eggs; *P* values for the comparison of unfertilized oocytes are in Source data. At least three biological replicates were performed for each experiment. Source data are available online for this figure.

To gain insight into BAF-1 localization, we knocked GFP and mCherry (mCh) cassettes into the endogenous *baf-1* and *baf-1(G12T)* loci to generate N-terminal fusions. We observed that BAF-1 is present in mature sperm, both in males (Appendix Fig. S1A) and hermaphrodites (Appendix Fig. S1B). In nematode sperm, chromosomes are surrounded by a poorly defined halo that presumably lacks nuclear membranes (Wolf et al, 1978). GFP::BAF-1(G12T) is also visible in sperm in hermaphrodites (Appendix Fig. S1C), indicating that BAF-1's localization is not affected by the NPGS mimicking mutation. In contrast to BAF-1, we could not detect endogenously tagged LMN-1 or EMR-1 in sperm of hermaphrodites (Appendix Fig. S1D), nor LMN-1 in sperm in males (Appendix Fig. S1E). Moreover, the *baf-1(G12T)* mutation did not induce localization of GFP::LMN-1 to sperm (Appendix Fig. S1F).

We next evaluated if the NGPS mutation affected lifespan, as would be expected for a progeria model. At 20 °C, *baf-1(G12T)* hermaphrodites lived as long as control animals (Fig. EV1D). However, because lifespan correlates inversely with fertility (Arantes-Oliveira et al, 2002; Berman and Kenyon, 2006), we speculated that the lower brood size in *baf-1(G12T)* at restrictive temperature might counteract a potential shortening of their lifespan. To address this, we first evaluated lifespan at 25 °C in the presence and absence of the transcription factor DAF-16, which is required to correlate fecundity with longevity (Hsin and Kenyon 1999). As expected, depletion of DAF-16 reduced lifespan in wild-type hermaphrodites, as well as in *baf-1(G12T)* ($p < 2e^{-16}$; Fig. 1E). Importantly, the *baf-1(G12T)* mutation caused a reduction in median lifespan at the restrictive temperature both in the presence (7% reduction; $p = 8e^{-13}$) and absence (9% reduction; $p = 7e^{-10}$) of DAF-16 (Fig. 1E). To completely eliminate the confounding effect of the germ line on lifespan, we ablated the germ line by using the *glp-4(ts)* mutation, which prevents germ cell proliferation at 25 °C (Beanan and Strome 1992). We found that median lifespan of sterile *glp-4(ts)* worms was reduced from 16 to 12 days (25% reduction; $p < 2e^{-16}$) by the *baf-1(G12T)* allele (Fig. 1F). These data indicate that the NGPS mutation affects fecundity and lifespan in a temperature-dependent manner.

## Accelerated age-induced nuclear morphology deterioration in *baf-1(G12T)* mutants

Alterations in NE morphology is associated with aging in several species, including *C. elegans* (Haithcock et al, 2005; Perez-Jimenez et al, 2014) and also in human progeria syndromes (Fragoso-Luna and Askjaer 2023). We therefore investigated morphology of nuclei

in hypodermal cells of wild-type and *baf-1(G12T)* animals at different ages. LMN-1 and EMR-1 endogenously tagged with GFP and mCherry (mCh), respectively, were used to visualize the NE at day 1, 6, and 8 of adulthood in worms raised at 25 °C. Nuclei were manually classified by single-blind observer based on their morphology as previously described (Perez-Jimenez et al, 2014), except that we introduced a fourth class to describe the most irregular nuclei (see Methods). Thus, while class I nuclei have smooth, uniform NE signal, classes II–IV represent progressive deterioration of nuclear morphology. At day 1, class I represented ~35% of nuclei from wild-type worms and only ~20% of nuclei from *baf-1(G12T)* mutants when evaluated with either GFP::LMN-1 or EMR-1::mCh (Fig. 2A–D; $p < 0.03$; Table EV1). At day 6, the number of class I nuclei in all strains was very reduced so we combined class I + II. Whereas class I + II represented ~90% of nuclei from wild-type worms at day 1, at day 6, this fraction was reduced to 27% according to GFP::LMN-1 (Fig. 2A,B) and 42% according to EMR-1::mCh (Fig. 2C,D). The slightly more deteriorated morphology in animals expressing GFP::LMN-1 compared to EMR-1::mCh suggested that tagging of LMN-1 affects it function, as described (Bone et al, 2016) (see also below). Regardless of the marker, significantly fewer class I and II nuclei and more class III and IV nuclei were observed in *baf-1(G12T)* on day 6. Scoring EMR-1::mCh, class I + II nuclei decreased from 42% in wild-type animals to 8% in *baf-1(G12T)* mutants (Fig. 2D; $p < 2.2e^{-16}$; Table EV1). Similarly, the frequency of class IV nuclei based on GFP::LMN-1 localization increased from 10% in control animals to 23% in *baf-1(G12T)* mutants (Fig. 2B; $p = 0.0004$; Table EV1). We also evaluated nuclear morphology after 8 days of adulthood at 25 °C, which confirmed an increased deterioration in *baf-1(G12T)* with both NE markers (Appendix Fig. S2; Table EV1). We concluded that alterations in nuclear morphology are significantly accelerated in *baf-1(G12T)* mutants at 25 °C.

## Nuclear envelope accumulation of lamin and emerin is reduced by the *baf-1(G12T)* mutation

The amino acid residue mutated in NGPS patients is positioned close to the interface between BAF and lamin (Samson et al, 2018) and experimental evidence indicates that the interaction between the two proteins is weakened by the mutation (Janssen et al, 2022). However, another study suggested that the interaction was unaffected (Paquet et al, 2014). Because the localization of BAF and lamins are interdependent (Margalit et al, 2005), impaired interaction could drive reduced BAF-1 and lamin accumulation at the NE. To test the impact of *baf-1(G12T)* on LMN-1, EMR-1, and

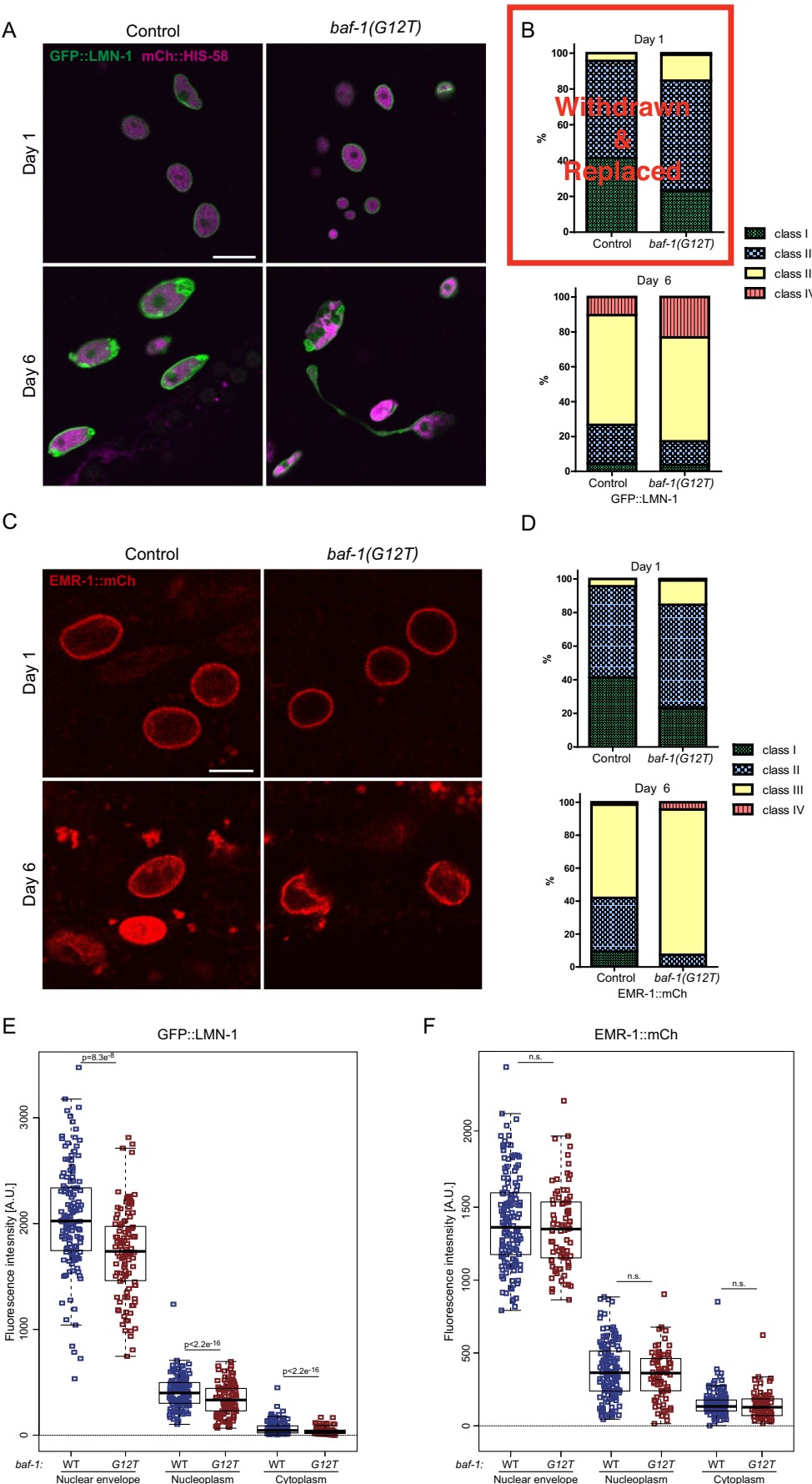

**Figure 2. Nuclear morphology deteriorates faster in *baf-1(G12T)* mutants.**

(A–D) Morphology of hypodermal nuclei was classified on day 1 and 6 of adulthood in strains expressing GFP::LMN-1 (green) and mCh::HIS-58 (magenta) (A, B) or EMR-1::mCh (C, D). Confocal micrographs (A, C) and combined data from 4 independent experiments (B, D) are shown. Scale bars represent 10 μm (A) or 5 μm (C). Classes I-IV are defined in Methods; briefly class I corresponds to nuclei with smooth, regular NE morphology and classes II–IV represent increasingly irregular morphologies. The number of nuclei analyzed, and statistical tests are reported in Table EV1. (E, F) Quantification of GFP::LMN-1 (E) and EMR-1::mCh (F) signal in the NE, nucleoplasm and cytoplasm of hypodermal cells. Data from 3 independent experiments; at least 15 animals were analyzed for each strain. In the boxplots, the midline represents the median and the upper and lower limits of the box correspond to the third and first quartile, respectively. The whiskers extend to the lowest and higher observations but maximum up to 1.5 times the interquartile range. *P* values from two-sided t-tests are indicated; *p* values ≥ 0.05 are considered not significant (n.s.). Source data are available online for this figure.

BAF-1 localization in vivo, we quantified these factors at the NE of hypodermal and intestinal cells. We observed significantly lower median GFP::LMN-1 signal at the NE in *baf-1(G12T)* mutants in both tissues at 20 °C and 25 °C (Figs. 2E and EV3A–C). In contrast, accumulation of EMR-1 at the NE was unaffected by the *baf-1(G12T)* mutation in both tissues at 25 °C and reduced in the hypodermis at 20 °C (Figs. 2F and EV3D–F). In human NGPS cells, emerin was observed to be delocalized to the ER (Puente et al, 2011; Janssen et al, 2022), but we detected no increase in cytoplasmic EMR-1::mCh signal in the mutant, indicating that this NGPS phenotype is not present in the *C. elegans* model. In agreement with these microscopy data, analysis of whole-worm mRNA levels by quantitative RT-PCR also revealed a significant reduction in *lmn-1* expression whereas *emr-1* was unaffected (Appendix Fig. S2E,F).

Surprisingly, we found that the combination of *baf-1(G12T)* with the endogenously tagged *gfp::lmn-1* allele *(yc32)* led to a fully penetrant maternal effect lethal (*mel*) phenotype: homozygous *lmn-1(yc32[gfp::lmn-1]); baf-1(G12T)* embryos produced by hermaphrodites heterozygous for the *yc32* allele developed all into adults that laid dead eggs at 20 °C (Fig. 3A). We ruled out the possibility that the lethality was due to an unknown mutation in the genetic background of the *yc32* allele by using a different endogenous *gfp::lmn-1* allele *(jf98)*, which also produced a 100% *mel* phenotype (Appendix Fig. S3A). In both cases, one copy of the wild-type *lmn-1* allele was sufficient to sustain normal development.

To determine the subcellular defects that could explain the lethality of the allelic combination *lmn-1(yc32[gfp::lmn-1]); baf-1(G12T)*, we recorded embryos by confocal 4D microscopy. Chromosomes were visualized by co-expression of mCh::HIS-58, which revealed segregation defects in early mitotic divisions and developmental arrest around 4-5 h post-fertilization (Appendix Fig. S3B). The early chromosome segregation defects prompted us to investigate the embryos with higher temporal resolution. This uncovered abnormal pronuclear appearance, incomplete sealing of the NE after mitosis and frequent chromosome bridges (6/10 embryos), which potentially is the primary cause to the observed lethality (Fig. 3B; Movie EV1). Moreover, cell divisions were delayed (compare AB anaphase and P1 metaphase at 1049 s in the control embryo versus 1149 s in the *baf-1(G12T)* embryo). These phenotypes are consistent with the effects of embryonic depletion of BAF-1 or LMN-1 (Liu et al, 2000; Margalit et al, 2005).

Next, we analyzed endogenously tagged *gfp::baf-1* and *gfp::baf-1(G12T)* nematodes. Strikingly, whereas >99% of eggs laid by homozygous *gfp::baf-1* hermaphrodites developed and hatched, all eggs laid by homozygous *gfp::baf-1(G12T)* hermaphrodites (derived from balanced, heterozygous hermaphrodites) were inviable (Fig. 3C). Similarly to the *lmn-1(yc32[gfp::lmn-1]); baf-1(G12T)*

embryos, the *gfp::baf-1(G12T)* embryos arrested before elongation (Fig. EV4A). Pronuclear appearance was severely affected and, in most embryos, the two pronuclei failed to juxtapose prior to the first division (Fig. 3D; Movie EV2; 4/7 embryos). Chromosome segregation and postmitotic nuclear reformation were abnormal in all *gfp::baf-1(G12T)* embryos, presumably being causative for the early developmental arrest. Cell division delays were also evident in all mutant embryos.

Finally, we compared GFP::BAF-1 and GFP::BAF-1(G12T) expression in adults and found that both proteins accumulated at the nuclear periphery in all somatic cells and in the germ line (Fig. EV4B). Quantifying the intensity at the NE or in the nucleoplasm of hypodermal cells did not demonstrate any difference between endogenously GFP-tagged wild-type and mutant BAF-1 (Fig. 3E). A small reduction in cytoplasmic signal was observed for BAF-1(G12T), however, no difference was detected in the ratio between nucleoplasmic/cytoplasmic signal (Fig. 3E). Quantitative RT-PCR analysis of whole-worm RNA samples also indicated that *baf-1* and *baf-1(G12T)* are expressed at identical levels (Appendix Fig. S2E,F). Taken together, our results are compatible with a model in which mutation of residue 12 of BAF reduces the affinity for lamin, leading to a mild reduction in the accumulation of lamin at the NE. Moreover, the mutation sensitizes BAF so tagging of either BAF-1(G12T) or lamin with GFP has detrimental effects on chromosome segregation and NE assembly.

## The DNA binding profile of BAF-1 is modified by the G12T mutation

BAF binds DNA in vitro without sequence specificity, and due to its location at the NE, BAF has been proposed to regulate chromatin organization (Zheng et al, 2000; Kind and van Steensel, 2014). Moreover, human BAF A12T has been described to have altered DNA binding capacity (Paquet et al, 2014), although others have disputed this claim (Marcelot et al, 2021). We decided to test whether wild-type and mutant BAF-1 have different chromatin association profiles. Because chromatin organization differs between tissues, subtle changes in chromatin association of BAF-1(G12T) might not be detectable if averaging across multiple tissues. We therefore performed tissue-specific DamID (DNA Adenine Methyltransferase Identification) by expressing trace amounts of either Dam::BAF-1 or Dam::BAF-1(G12T) in two different tissues using a FLP-FRT system (Munoz-Jimenez et al, 2017; Cabianca et al, 2019). We focused on the hypodermis and the intestine because of skin atrophy and decreased subcutaneous fat in NGPS patients (Cabanillas et al, 2011; Fisher et al, 2020).

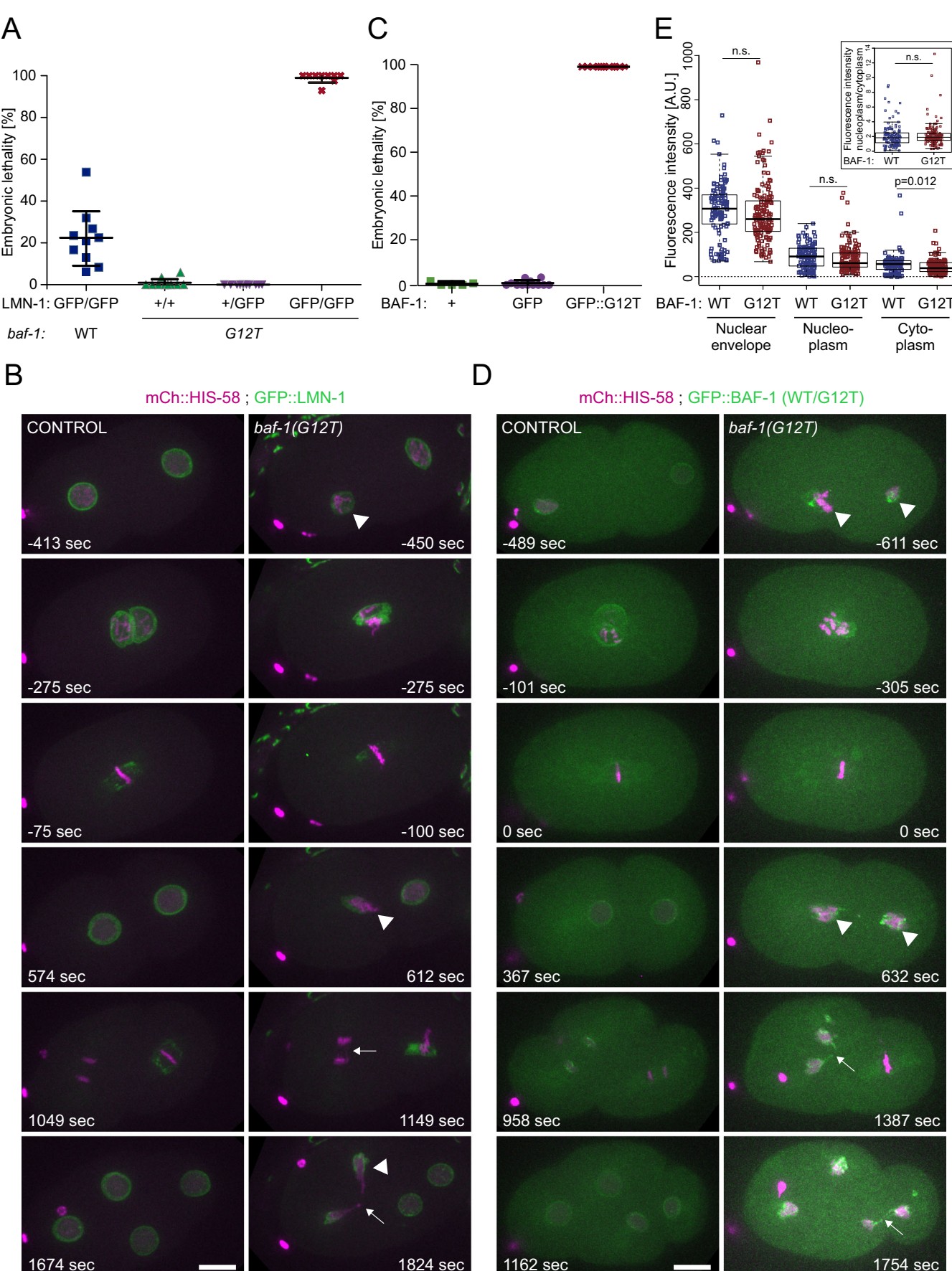

**Figure 3. *baf-1(G12T)* worms are hypersensitive to GFP tagging.**

(A) Embryonic lethality was quantified for hermaphrodites carrying combinations of *baf-1(G12T)* and *gfp::lmn-1(yc32)* and wild-type alleles as indicated. (B) Selected time points from confocal time-lapse recordings of early embryos expressing GFP::LMN-1 (green) and mCh::HIS-58 (magenta). (C) Lethality of embryos produced by either wild-type (+), GFP::BAF-1 (GFP) or GFP::BAF-1(G12T) (GFP::G12T) hermaphrodites. Each symbol in (A) and (C) represents the average per 3–5 worms. Wide lines and error bars represent mean and standard deviation, respectively. (D) Selected time points from confocal time-lapse recordings of early embryos expressing either BAF-1 or BAF-1(G12T) tagged endogenously with GFP. Arrowheads and arrows in (B) and (D) indicate abnormal (pro-)NEs and failures in chromosome segregation, respectively. Time is indicated relative to P0 anaphase onset. Scale bars represent 10 μm. (E) Quantification of GFP::BAF-1 (blue; $n = 9$ adult hermaphrodites) and GFP::BAF-1(G12T) (red; $n = 7$) intensity at the NE of hypodermal nuclei, in the nucleoplasm and in the cytoplasm. The ratio between the nuclear and cytoplasmic compartment is shown in the insert. In the boxplots, the midline represents the median and the upper and lower limits of the box correspond to the third and first quartile, respectively. The whiskers extend to the lowest and higher observations but maximum up to 1.5 times the interquartile range. P value from two-sided t-test is indicated; p values ≥ 0.05 are considered not significant (n.s.). Data from 2 (A, C) or ≥3 (B, D, E) independent experiments. Source data are available online for this figure.

DamID relies on the expression of the protein of interest fused to the DNA adenine methyltransferase (Dam) of *Escherichia coli*. When the protein of interest binds chromatin, Dam methylates GATC sites in close vicinity, which subsequently are identified through a series of enzymatic reactions and deep sequencing (de la Cruz Ruiz et al, 2022). To normalize for variations in chromatin accessibility, standard DamID protocols include a parallel sample based on expression of Dam alone or Dam fused to a protein with no affinity for chromatin, such as GFP. Thus, DamID can provide information about both chromatin accessibility and specific binding profiles in the same experiment.

We first determined chromatin accessibility in L4 larvae from biological triplicates by binning reads to 2 kb bins (Appendix Fig. S4A,B). We identified 1924 bins with increased accessibility in the hypodermis versus the intestine (3.8% of genome; FDR < 0.05) and 174 bins with increased accessibility in the intestine (0.35% of genome). Examples of differential accessibility include the genes *mua-6* and *dod-3*, which are expressed and more accessible in hypodermis and intestine, respectively (Fig. 4A). In contrast, we did not find evidence for alterations in chromatin accessibility when comparing wild-type and *baf-1(G12T)* nematodes and correlations were higher between the two genotypes within each replica than among replicas of the same genotype (Appendix Fig. S4A). This suggests that BAF-1 does not regulate compaction of interphase chromatin or that the mutation of residue 12 does not affect this function.

We next calculated the log2 ratios between either Dam::BAF-1 or Dam::BAF-1(G12T) and GFP::Dam to determine the binding profiles of BAF-1 and BAF-1(G12T). To visualize the global distribution of the two proteins, the genome was binned into 100 kb windows. This revealed an arm-center-arm pattern, where frequent association to BAF-1 or BAF-1(G12T) was detected in the distal thirds of all autosomes but not in the central parts (Appendix Fig. S5B). The pattern was highly similar between the two tissues, with few exceptions, such as the left arms of chrIII and chrIV. Comparing the log2 values across all samples showed that the two tissues are more dissimilar than BAF-1 and BAF-1(G12T) within each tissue (Appendix Fig. S5A). The arm-center-arm pattern has been reported for other NE proteins, including LMN-1, EMR-1 and LEM-2 at different developmental stages (Ikegami et al, 2010; Towbin et al, 2012; Gonzalez-Aguilera et al, 2014), as well as for the heterochromatin mark H3K9me (histone H3 methylated on lysine 9) (Gerstein et al, 2010).

By increasing the resolution to 10 kb bins, we observed more continuous interaction of BAF-1 and BAF-1(G12T) on autosome arms in hypodermis versus intestine (Fig. 4B,C). The G12T

mutation produced a small increase in association in chromosome centers specifically in the hypodermis. We next identified bins significantly enriched for BAF-1 or BAF-1(G12T) association and selected those for which the difference in log2 value between BAF-1 and BAF-1(G12T) was >0.58 (equal to a fold change of 1.5). For BAF-1 in hypodermis, 410 bins on autosomes (5%) fulfilled this criterium and the vast majority (87%) were located in arms (Table EV2; $p < 2.2e^{-16}$). In contrast, for BAF-1(G12T) in hypodermis, enrichment was mostly observed in chromosome centers (Table EV2; 478 enriched bins; $p = 5.0e^{-10}$). More bins were differentially associated with BAF-1 or BAF-1(G12T) in intestine compared to hypodermis, but they were more evenly distributed. We concluded that both BAF-1 and BAF-1(G12T) bound predominantly to heterochromatin-enriched chromosome arms and that regions with subtle but significant association to BAF-1(G12T) were also detected in chromosome centers in the hypodermis.

## BAF-1(G12T) differential gene association

Analysis at 2 kb bin and "gene level" (see Methods) enabled identification of genes bound in the two tissues. Genome browser views illustrated again higher similarity between BAF-1 and BAF-1(G12T) within tissues than either protein between tissues (Appendix Fig. S5C). Nevertheless, many bins were predominantly bound by either wild-type or mutant BAF-1 (Fig. 4D; Appendix Fig. S5C). In the hypodermis, we identified 5467 protein coding and non-coding genes bound by BAF-1 and 6376 genes bound by BAF-1(G12T) (Dataset EV1). 4541 genes (62%) associated with both proteins, whereas 2761 genes (38%) are specific to either wild-type (926) or mutant (1835) BAF-1 (Fig. 4E). In the intestine, 8442 genes and 7412 genes were bound by BAF-1 and BAF-1(G12T), respectively. Of these, 5410 genes (51%) were identified in both datasets, whereas 3032 genes (29%) associated specifically with BAF-1 and 2078 genes (20%) specifically with BAF-1(G12T) (Fig. 4E). Comparing all 4 datasets, 8056 genes (62%) are present in at least two datasets whereas 4839 genes (38%) are unique; 2552 genes (20%) are bound by wild-type and mutant BAF-1 in both tissues (Dataset EV1).

Analyzing the BAF-1 and BAF-1(G12T)-bound genes for tissue enrichment (Angeles-Albores et al, 2016) retrieved the categories *Nervous system*, *Epithelial system*, *Male*, and *Amphid sheath cell* in both hypodermis and intestine and by both wild-type and mutant BAF-1 (Fig. 4F; Dataset EV1). In contrast, the categories *Reproductive system*, *Germ line*, and *Male distal tip* were enriched for binding to wild-type BAF-1, but not BAF-1(G12T) in the

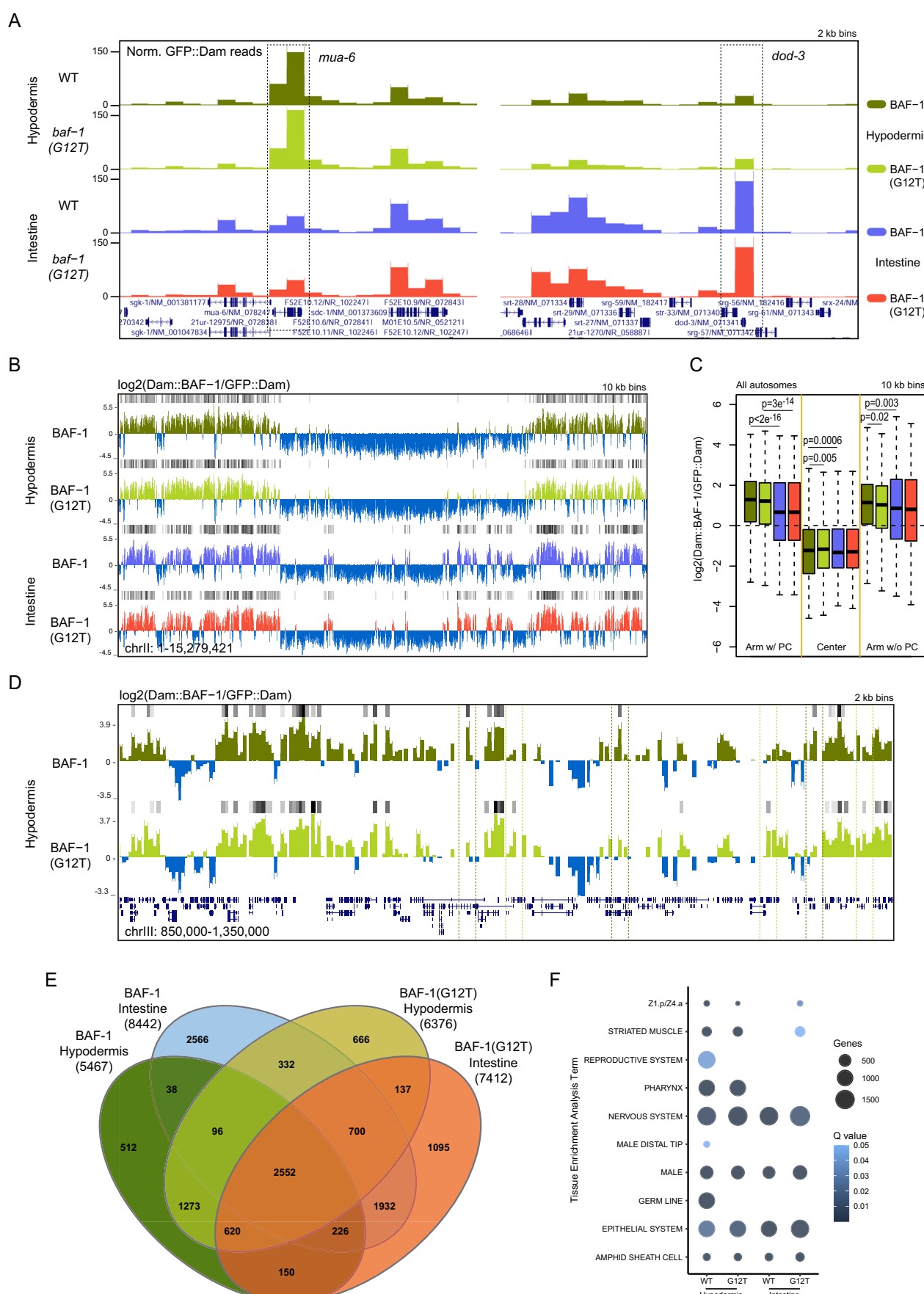

◄ **Figure 4.  Identification of tissue-specific BAF-1 and BAF-1(G12T) chromatin association.**

(A) Genome browser views of normalized GFP::Dam reads per 2 kb bins averaged across 3 independent biological replicas. The genes *mua-6* and *dod-3*, which are expressed in hypodermis and intestine, respectively, are indicated. (B, C) Normalized log2 values of Dam::BAF-1/GFP::Dam and Dam::BAF-1(G12T)/GFP::Dam ratios per 10 kb bins shown as genome browser view of chrII (B) and as boxplot for all autosomes (C). Gray bars above each track indicate enriched association to BAF-1 or BAF-1(G12T) (FDR < 0.05). Chromosome arms were grouped based on the presence or absence of meiotic pairing centers (PCs). Each group contains >2000 10 kb bins. In the boxplot, the midline represents the median and the upper and lower limits of the box correspond to the third and first quartile, respectively. The whiskers extend to the lowest and higher observations but maximum up to 1.5 times the interquartile range. For clarity, only selected *p* values from pairwise Wilcoxon Rank Sum Tests corrected for multiple testing (Benjamini and Hochberg method) are indicated. (D) Normalized log2 values of Dam::BAF-1/GFP::Dam and Dam::BAF-1(G12T)/GFP::Dam ratios per 2 kb bins shown as genome browser view of 500 kb region from left arm of chrIII. Examples of regions enriched for association to either BAF-1 or BAF-1(G12T) in the hypodermis are outlines with dashed boxed. (E, F) Venn diagram representation (E) and Tissue Enrichment Analysis (F) of genes associated with BAF-1 or BAF-1(G12T) in hypodermis or intestine. Total number of genes for each condition is shown in bracket in (E). The size and color of the circles in F reflect number of genes and *p* values adjusted for multiple comparisons (Benjamini and Hochberg method; termed Q value and determined using WormBase's Enrichment Analysis as described in Methods and protocols), respectively. Source data are available online for this figure.

hypodermis (Fig. 4F). For all these categories, except *Epithelial system*, genes are generally inactive in the two tissues that we analyzed. This suggested that BAF-1 binding correlates inversely with gene expression (see Discussion regarding the *Epithelial system*).

## Deregulation of ribosomal and cuticle genes in *baf-1(G12T)* mutants

We next used tissue-specific RNA polymerase DamID (RAPID) (Gomez-Saldivar et al, 2020; Fragoso-Luna et al, 2023) to identify genes expressed in hypodermis and intestine in wild-type and *baf-1(G12T)* mutants. In wild-type animals, 1960 genes were detected as expressed in hypodermis and 1585 genes in intestine (Dataset EV2). Of these, 491 genes were shared between the two tissues. To explore if association to BAF-1 correlates with expression, we calculated the median and average log2(Dam::BAF-1/GFP::Dam) values for genes expressed in both tissues, in only one of them or in neither. In the hypodermis, we found that the log2 values were lowest for genes expressed in this tissue, whereas genes expressed in intestine had intermediate values (Fig. 5A). A similar picture emerged when analyzing genes expressed only intestine (Fig. 5B). These data suggest that expressed genes are generally depleted for interaction with BAF-1.

By comparing hypodermal RAPID scores between wild-type and *baf-1(G12T)* we identified 36 genes that were reproducibly upregulated in the mutant and 26 genes that were downregulated (Fig. 5C; Dataset EV2). More genes were deregulated in the intestine with 76 genes being more expressed in *baf-1(G12T)* and 53 genes being repressed (Fig. 5D; Dataset EV2). Although only 0.9–2.5% of expressed genes in the two tissues were deregulated, gene ontology (GO) analysis revealed a significant overrepresentation of several GO terms. Genes related to histone acetylation, proton transport and ribosomes were deregulated in both tissues, whereas other classes appeared only in hypodermis (e.g., cuticle components) or intestine (Fig. 5E).

Although we found that expressed genes are generally devoid of interaction with BAF-1, we speculated that the deregulation of the genes identified in our RAPID experiments could be related to altered BAF-1 binding. We plotted the DamID scores obtained with Dam::BAF-1(G12T) versus Dam::BAF-1 for the deregulated genes, focusing initially on the intestine because of the higher number of deregulated genes. The majority of deregulated genes had negative log2 scores, implying more frequent interaction with GFP::Dam

than with either Dam::BAF-1 or BAF-1(G12T). Surprisingly, both upregulated (Fig. 6A) and downregulated (Fig. 6B) genes had higher log2 scores for BAF-1(G12T) compared to the wild-type protein. This result indicates that deregulated genes are bound more often when BAF-1 it mutated. Two exceptions appeared among the downregulated genes: F14E5.8 (an uncharacterized protein with no obvious ortholog in vertebrates) and *emr-1*, which both were bound by BAF-1 but not BAF-1(G12T) (Fig. 6B). Whether BAF-1 is directly involved in transcriptional activation of these genes remains to be tested. As described above, the amount of endogenously tagged EMR-1::mCh at the NE of intestinal cells was normal in *baf-1(G12T)* mutants (Fig. EV3F), suggesting a cellular capacity to buffer the downregulation of *emr-1* transcription (Vogel and Marcotte, 2012). We also noted that among the 129 deregulated genes in the intestine, 13 encode ribosomal proteins (Q value = $2.4e^{-9}$). Strikingly, all 5 upregulated and 7/8 downregulated ribosomal genes had higher log2 scores with BAF-1(G12T) than with BAF-1 (Fig. 6A–C; *p* = 0.0009). The genes that were deregulated in the hypodermis showed little difference in their interaction with the two versions of BAF-1 (Fig. EV5A, B). However, genes encoding cuticle constituents were overrepresented (8/62 genes, Q value = $1.2e^{-6}$) and 7 of these showed decreased interaction with BAF-1(G12T) compared to BAF-1 (Fig. EV5C; *p* = 0.03). Thus, although there is no uniform correlation between changes in association to BAF-1 and changes in gene expression, several gene classes appear to have specific behaviors.

## BAF-1(G12T) induces alterations in stress resistance

Cells derived from HGPS patients suffer chronic oxidative stress and stimulation of antioxidant pathways can reverse cellular HGPS phenotypes (Kubben et al, 2016). Because of the similarities between NGPS and HGPS, we speculated that *baf-1(G12T)* mutants might be hypersensitive to oxidative stress. Surprisingly, we observed that *baf-1(G12T)* mutants survived longer than control animals upon exposure to tert-butyl hydroperoxide (Fig. 7A; *p* < $2e^{-16}$). Based on this result, we tested if the *baf-1(G12T)* mutants displayed changes in resistance to other systemic stressors. Gene ontology analysis suggested an enrichment in genes involved in Response to X-ray in baf-1(G12T) mutants. We therefore exposed young adults to DNA damaging ultraviolet light and scored survival in the progeny. *baf-1(G12T)* mutants exhibited reduced survival in 7/8 samples (Fig. 7B; *p* = 0.04), indicating that the ability to repair UV-induced DNA damage is compromised by the NGPS

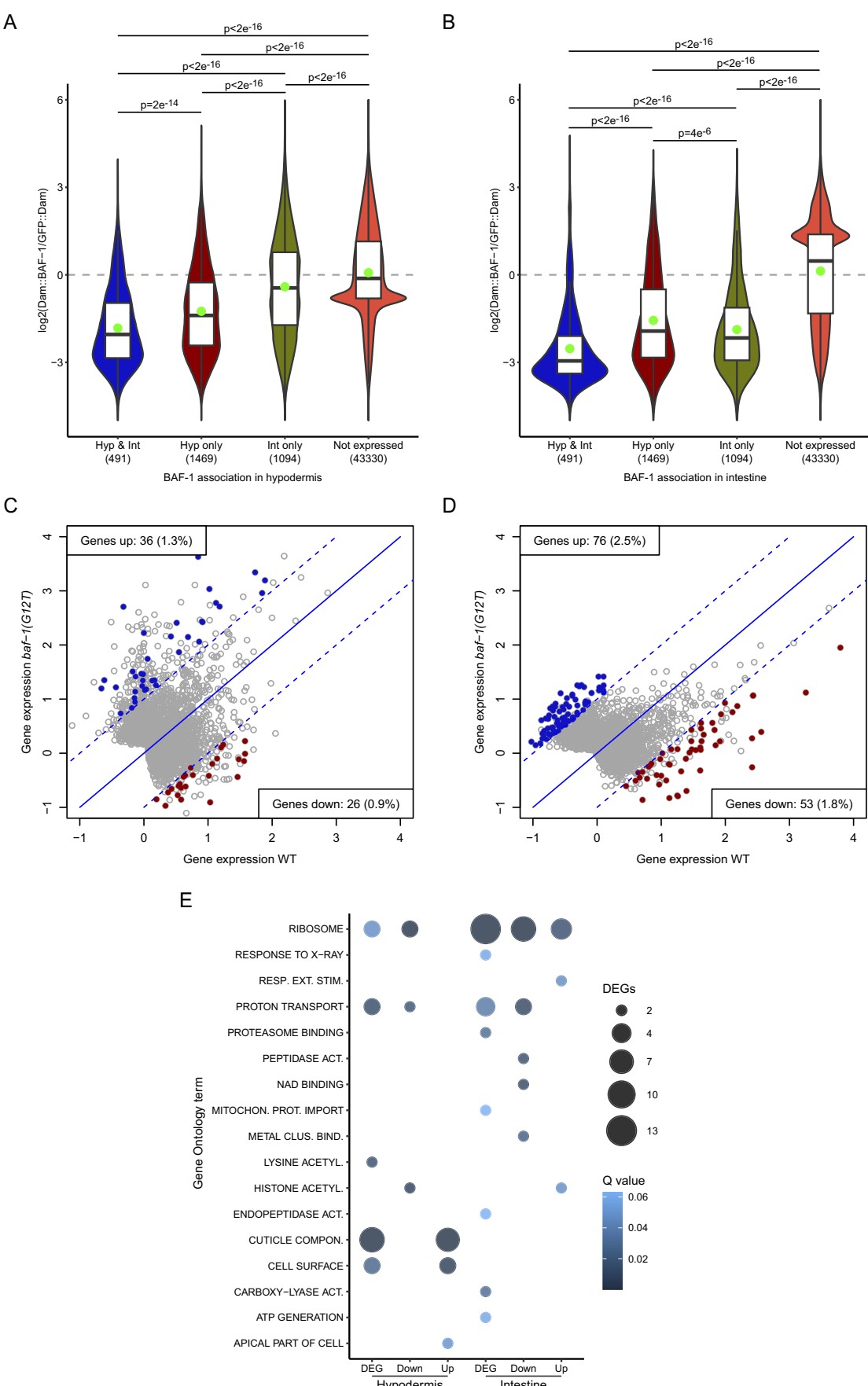

◄

**Figure 5.   Identification of differentially expressed genes.**

(A, B) Distribution of log2 values of Dam::BAF-1/GFP::Dam and Dam::BAF-1(G12T)/GFP::Dam ratios per gene in hypodermis (A) and intestine (B). Genes were separated in 4 categories: expressed in hypodermis and intestine (Hyp & Int; 491 genes), hypodermis only (Hyp only; 1469 genes), intestine only (Int only; 1094) or not detected as expressed by RAPID in any of the two tissues (Not expressed; 43,330 genes). The midlines of the boxes represent the median and the upper and lower limits of the box correspond to the third and first quartile, respectively. P values from pairwise Wilcoxon Rank Sum Tests corrected for multiple testing (Benjamini and Hochberg method) are indicated. (C, D) Gene expression determined by RAPID in wild-type (x-axis) versus *baf-1(G12T)* (y axis) in hypodermis (C; n = 2759 genes) and intestine (D; n = 2999 genes). Genes outside the dashed lines have average differential log2 values >1 between the wild-type and the mutant (fold change >2). Genes detected as differentially expressed in both replicas are indicated in blue (Genes up in *baf-1(G12T)*) and red (Genes down in *baf-1(G12T)*). (E) Significant gene ontology terms identified among differentially expressed genes (DEG), all combined, but also upregulated and downregulated separately. The size and color of the circles reflect the number of DEGs and p values adjusted for multiple comparisons (Benjamini and Hochberg method; termed Q value and determined using WormBase's Enrichment Analysis as described in Methods and protocols), respectively.

mutation. We next tested the response to acute heat stress, which provokes protein misfolding and aggregation and consequently activation of several response pathways, such as heat shock response, unfolded protein response and autophagy. When incubated at 35 °C, the median survival of control hermaphrodites was 6 h versus 5 h for *baf-1(G12T)* mutants (Fig. 7C; p = 0.00001). Thus, mutation of BAF-1 led to significantly reduced heat stress resistance and further studies should determine which response pathway(s) is impaired. We concluded that the NGPS mutation causes increased sensitivity to certain stress factors, but also that the mutation can protect against oxidative stress, which contrasts the situation in HGPS.

## Discussion

The need for a deeper understanding of healthy aging is becoming increasingly important, as the life expectancy of the world population has augmented in the last century. Studies on accelerating aging diseases have provided insight on different pathways that are altered during physiological aging, but specific mechanisms underlying these conditions have also emerged. The use of animal models is an instrumental source of knowledge due to the possibility of studying the effect of mutations and interventions at multiple scales. In this study, we established *C. elegans* as a model to study how a progeria-causing mutation in BAF affects nuclear organization, gene expression and physiology. Our work has provided insight not only to the differences in chromatin association between wild-type and mutant BAF-1 but also on tissue-specific interactions of BAF-1 with predominantly silent genes. The observation that nuclear morphology deteriorated faster in *baf-1(G12T)* mutants argues that *C. elegans* is a relevant model to investigate NGPS. Importantly, the differential sensitivities to irradiation, temperature and oxidative stress suggest a complex organismal response to the G12T substitution in BAF-1 and warrant further investigation. Recently, *Drosophila* was reported as another invertebrate NGPS model, which revealed defects in centromere function and ovary homeostasis (Duan et al, 2023). We anticipate that these two relatively simple animal models will provide important platforms for both discovery and validation of potential targets and treatments.

NGPS patient-derived fibroblasts exhibits partial delocalization of emerin from the NE to the cytoplasm (Puente et al, 2011; Janssen et al, 2022). Although we observed a mild but significant decrease in NE accumulation of EMR-1° in the hypodermis of *baf-1(G12T)* mutants at 20 °C, we did not see an increase in cytoplasmic EMR-1 levels. The

behavior of emerin in the *Drosophila* NGPS model (Duan et al, 2023) was like the situation in *C. elegans*, suggesting that emerin might be affected by the NGPS mutation in a species and/or cell type-specific manner, for instance due to the presence or absence of other interaction partners. NE accumulation of LMN-1 was reduced in *baf-1(G12T)* mutants, which is compatible with a model in which 1) lamin depends on BAF to localize at the NE (Margalit et al, 2005) and 2) the affinity of BAF for lamin is reduced by the NGPS mutation (Samson et al, 2018; Janssen et al, 2022). We found that localization of BAF-1 itself was unaffected in the NE of *C. elegans* somatic and germline tissues. This result contrasts with observations in flies, where the NGPS mutation led to reduced NE accumulation (Duan et al, 2023). Different behaviors of BAF A12T in human cells have been reported: ectopic expression showed in one study no difference in the accumulation of wild-type and mutant BAF at the NE (Paquet et al, 2014), whereas another study concluded that NE accumulation of BAF A12T was impaired (Puente et al, 2011). Finally, using antibodies to compare endogenous BAF localization in control and NGPS fibroblasts found a reduced nucleocytoplasmic ratio in the latter (Janssen et al, 2022).

Tagging BAF-1 and BAF-1(G12T) endogenously with GFP revealed a striking difference during early development. Whereas tagging of wild-type BAF-1 had no obvious effect on fitness at any stage of development, *gfp::baf-1(G12T)* embryos suffered severe nuclear appearance and chromosome segregation defects already from the 1 cell-stage and all died before hatching. This was in stark contrast with the ~100% viability of untagged *baf-1(G12T)* embryos. Surprisingly, endogenous GFP tagging of LMN-1 also caused a fully penetrant embryonic lethal phenotype when combined with the *baf-1(G12T)* allele. This lethality seemed specific to LMN-1 because *emr-1::mCh; baf-1(G12T)* individuals developed normally. Moreover, time-lapse recordings unraveled similar defects in early *gfp::baf-1(G12T)* and *gfp::lmn-1; baf-1(G12T)* embryos. One possible explanation is that GFP physically interferes with the interaction between BAF-1 and LMN-1 and that this is only critical when BAF-1 carries the G12T mutation. Steric hindrance can be envisioned both in the case of N-terminal tagging of BAF-1 because of proximity of BAF's N terminus to the BAF·lamin interface, and in the case of N-terminal tagging of LMN-1 because lamin dimers form head-to-tail polymers where the BAF-interacting globular domain of lamin might be in proximity with the N terminus of the neighboring lamin molecule (Turgay et al, 2017; Samson et al, 2018). In support of this hypothesis, animals with endogenously tagged LMN-1 and EMR-1 (*gfp::lmn-1; emr-1::mCh*) were fully viable, whereas we could not create a strain that was homozygous for endogenous GFP::LMN-1 and mCh::BAF-1.

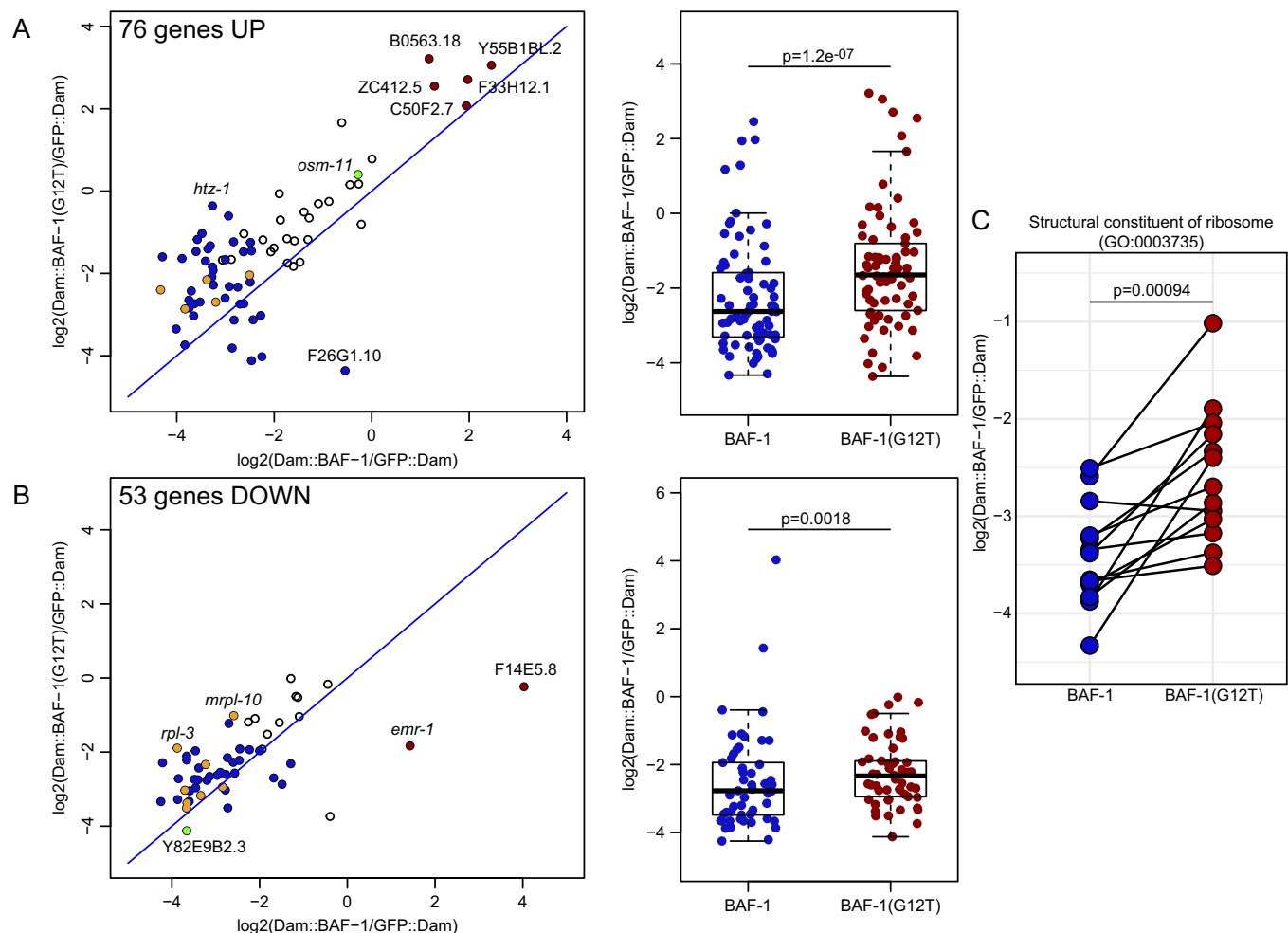

**Figure 6. Differentially expressed genes bind distinctively to BAF-1 and BAF-1(G12T).**

(A, B) Genes with increased (A; $n = 76$) or decreased (B; $n = 53$) expression in intestine of *baf-1(G12T)* mutants were analyzed for association to wild-type BAF-1 and BAF-1(G12T) in scatter plots (left; BAF-1 association on x axis; BAF-1(G12T) association on y axis) and boxplots (right). Genes with significantly higher association to Dam::BAF-1 or Dam::BAF-1(G12T) than to GFP::Dam are indicated in red in scatter plots. Genes indicated in orange encode structural constituents of ribosomes. Genes plotted in green are also deregulated in hypodermis (see Fig. EV5). In the boxplots, the midline represents the median and the upper and lower limits of the box correspond to the third and first quartile, respectively. The whiskers extend to the lowest and higher observations but maximum up to 1.5 times the interquartile range. (C) The association of differentially expressed genes encoding structural constituents of ribosomes to BAF-1 and BAF-1(G12T) in intestine ($n = 13$). P values from two-sided paired t-tests, excluding outliers in (A, B), are indicated.

We conclude that insertion of fluorescent tags has a dramatic effect in *baf-1(G12T)* nematodes as well as in flies (Duan et al, 2023). Although further studies are needed to determine the molecular mechanism, we note that the high lethality of *gfp::lmn-1; baf-1(G12T)* embryos provides an ideal sensitized genetic background to screen for suppressors (Breusegem et al, 2022).

Tissue-specific DamID unveiled that BAF-1 bound predominantly to the distal arms of chromosomes, which are characterized by accumulation of repetitive elements and heterochromatin marks, in particular methylation of histone H3 on lysine 9 (Gerstein et al, 2010; Ahringer and Gasser, 2018). In accordance with BAF-1's direct interactions with LMN-1, EMR-1 and LEM-2, DamID and chromatin immunoprecipitation profiles of these three proteins also show enrichment in chromosome arms (Ikegami et al, 2010; Towbin et al, 2012; Gonzalez-Aguilera et al, 2014). Although the profiles of BAF-1 in hypodermis and intestine overlapped closely throughout most of the genome, we also detected regions larger than 100 kb that only interacted with BAF-1 in one of the tissues. At the level of individual genes, we found that expressed genes identified in our tissue-specific RAPID experiments were largely devoid of interaction with BAF-1. Although this was expected, based on the trend that genes positioned at the nuclear periphery are often transcriptionally silent (Ahringer and Gasser 2018; Manzo et al, 2022), our study represents the first mapping of BAF binding sites at gene resolution in any species. We observed an enrichment of BAF-1 at genes related to the nervous system and male-specific structures both in hypodermis and intestine where these genes are mostly silent. The appearance of the category *Epithelial system* as enriched for BAF-1 binding in hypodermis and intestine was surprising because we expected genes in this category to be expressed in these tissues. However, *Epithelial system* encompasses several thousand genes and inspection of the BAF-1-bound genes in

A

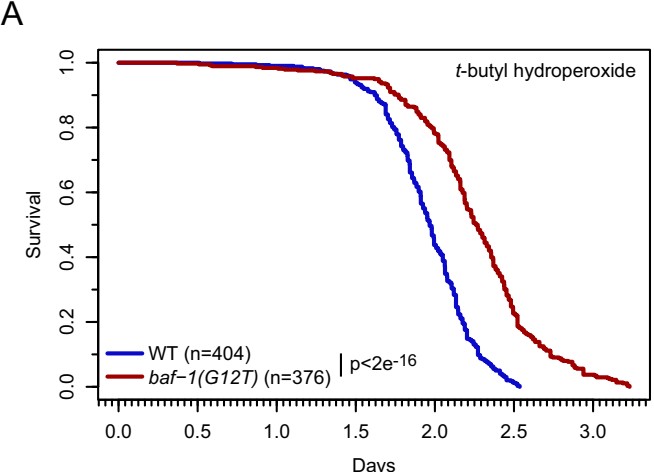

**Figure 7. Altered stress resistance on *baf-1(G12T)* mutants.**

(**A**) Survival of young adult hermaphrodites as function of exposure time to 6 mM tert-Butyl hydroperoxide. Numbers in brackets indicate the number of animals analyzed. (**B**) Percentage of embryos developing into larvae after exposure of young adult hermaphrodites to either 0 (left) or 25 (right) J/m² dose of UV light. Each point represents an independent replica (*n* = 8). (**C**) Survival of young adult hermaphrodites as function of incubation time at 35 °C. Numbers in brackets indicate the total number of animals analyzed in three independent replicas. *P* values from Chi-squared tests (**A**, **C**) and two-sided paired t-test (**B**) are indicated.

B

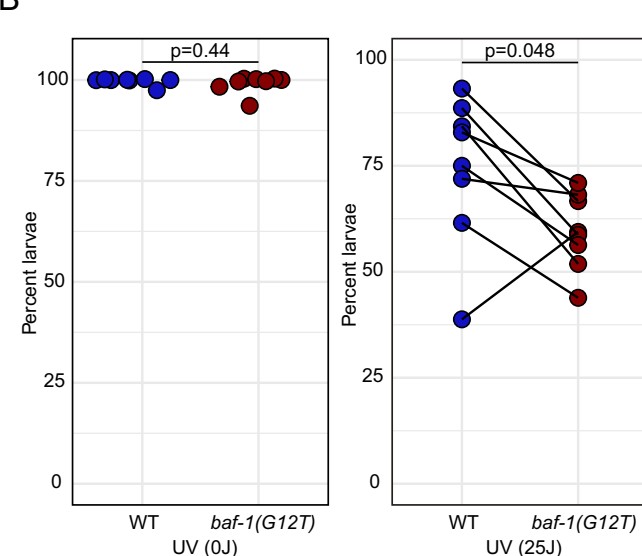

C

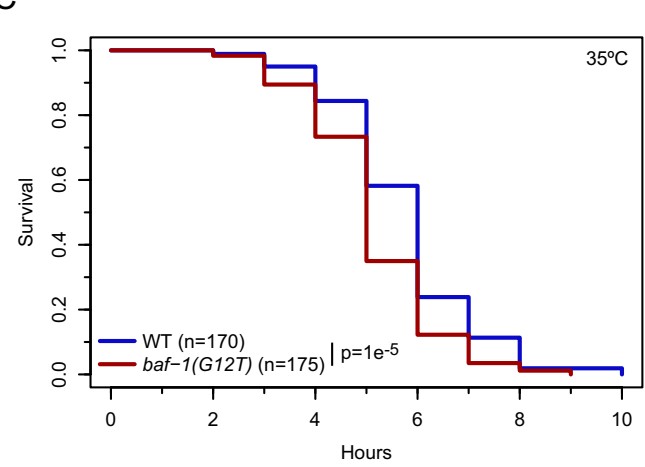

this category revealed that <3% were detected as expressed in the hypodermis in our RAPID experiments or in another recent hypodermal transcriptomic dataset (Katsanos et al, 2021). We therefore concluded that BAF-1 binding in general correlates inversely with gene expression.

On a global scale, the chromatin association profile of BAF-1 and BAF-1(G12T) were highly similar, indicating that the NGPS mutation does not provoke chromosome repositioning. However, many changes were observed at the level of a few kb. In each of the two tissues, BAF-1 and BAF-1(G12T) bound 7302–10,520 protein coding and non-coding genes (16–23% of all genes). More than half of the genes (51–62%) were shared between the wild-type and the mutant protein, whereas the number of uniquely bound genes ranged from 926 to 3032 (13–29%). In the hypodermis, the different binding profiles manifested in several tissue enrichment analysis terms appearing only with wild-type BAF-1 (*Reproductive system*, *Germ line*, and *Male distal tip*). Despite the large number of genes that showed differences in association to BAF-1 versus BAF-1(G12T), relatively few genes were detected as deregulated. DamID-based methods, such as RAPID and TaDa, use RNA polymerase occupancy as a reliable proxy for gene expression (Southall et al, 2013; Gomez-Saldivar et al, 2020; Katsanos et al, 2021). However, subtle changes in transcription might not be detectable. Among the deregulated genes, several were related to metabolic processes (e.g., *Proton transport, ATP generation, Mitochondrial protein import*, etc.) and chromatin regulation (*Histone acetylation*). The most frequent deregulated genes were *Cuticle components* in the hypodermis and *Ribosome components* in both tissues. Seven of the eight deregulated cuticle genes in the hypodermis associated less frequently with BAF-1(G12T) than with BAF-1. For ribosomal genes the correlation was even more pronounced and in the opposite direction: 12 of the 13 deregulated ribosomal genes in the intestine associated less frequently with BAF-1(G12T) than with BAF-1. Surprisingly, in the intestine, both upregulated and downregulated genes bound more frequently to the mutant BAF-1 protein. The genes that changed both association with BAF-1 and expression level in the context of the G12T mutation might represent primary NGPS alterations. Importantly, translation levels and fidelity were observed to be altered in NGPS fibroblasts and depletion of ribosome components could improve several cellular parameters (Breusegem et al, 2022). At current, we can only speculate about how the NGPS mutation might affect gene expression. Proteomics analyses indicate that BAF interacts with several histones and transcription factors (Montes de Oca et al, 2009), and the differences between BAF-1 and BAF-1(G12T)'s chromatin-binding profiles reported here might be accompanied by changes in the association of chromatin factors at the deregulated

loci. A particularly interesting candidate is GCL-1/germ cell-less 1, a repressive factor involved in spermatogenesis (Holaska et al, 2003). Moreover, it is plausible that the diminished recruitment of LMN-1 to the NE in *baf-1(G12T)* mutants modifies its interaction with the genome and with chromatin factors.

Limited information is available on fertility in progeria patients (Donadille et al, 2013), but mutations in components of the NE LINC (linker of nucleoskeleton and cytoskeleton) complex cause infertility in several animal species (Link and Jantsch, 2019). We observed a strong reduction in brood size of *baf-1(G12T)* hermaphrodites at 25 °C, which at least in part could be ascribed to a diminished presence of sperm in the spermatheca. Sperm production in hermaphrodites occurs during the L4 stage. The observation that the brood size phenotype depended on incubation at 25 °C during L4 stage or before points towards sperm-related defects. Moreover, more than one-third of the oocytes that passed through the spermatheca of *baf-1(G12T)* hermaphrodites on day 1 of adulthood were not fertilized, despite sperm still being present. This fertility defect was rescued by mating with wild-type males, suggesting that *baf-1(G12T)* sperm were abnormal but oocytes unaffected. Mating feminized worms to *baf-1(G12T)* males produced significantly fewer progeny compared to *fog-2* males, suggesting *baf-1(G12T)* also causes defective male in sperm. Sperm, in contrast to oocytes, lack a typical nuclear membrane. Instead, a halo on undefined composition encapsulates sperm DNA, RNA, and centrioles (Wolf et al, 1978). In this study, we found that BAF-1 but neither LMN-1 nor EMR-1 was present in mature sperm in adults of both sexes. Moreover, the *baf-1(G12T)* mutation did not alter this behavior. It is therefore possible that BAF-1 has a specific role in spermatogenesis that is affected by the NGPS mutation, but further studies are required to explore this.

Stress resistance correlates positively with longevity, whereas alterations in stress response pathways are intrinsically linked to diseases (Soo et al, 2023). In this context, we consider the reduced resistance of *baf-1(G12T)* mutants to temperature stress and UV irradiation reported in this study as a reflection of an overall impaired health span. We envision that they can be explored as traits in future experiments designed to identify genes or compounds that improve the health of *baf-1(G12T)* nematodes. However, the *baf-1(G12T)* mutation increased resistance to oxidative stress, highlighting the complex interplay between different stress response pathways and the need to evaluate potential interventions in multiple conditions to minimize negative side effects. In this aspect, invertebrate models, like nematodes and flies, are very attractive due to easy manipulation, low cost, and fast generation times (Yamamoto et al, 2024). We note that the simplicity of invertebrates also implies certain limitations. For instance, while both human and *C. elegans* genomes contain a single BAF gene, humans, but not *C. elegans*, express multiple lamin isoforms in tissue-specific ratios that regulate chromatin organization and nuclear mechanics (Swift et al, 2013). Thus, *C. elegans* is not suitable to explore potential differences in how wild type and NGPS BAF interacts differently with the various lamin isoforms.

In conclusion, we have characterized the effect of introducing in *C. elegans* a mutation described to cause NGPS in humans. Our results support the proposal that the NGPS mutation induces changes in BAF's interaction with lamin and they are also consistent with the hypothesis that the mutation alters BAF's affinity for chromatin. Importantly, *baf-1(G12T)* mutants reproduce several progeria phenotypes, including reduced lifespan, accelerated nuclear deterioration, lower fertility and altered stress resistance. We propose that these characteristics make *C. elegans* a relevant model to study the molecular mechanisms of NGPS and to explore possible interventions. Hopefully, combining the advantages of a simple in vivo model with those of human cell and organoid cultures will lead to the discovery of efficient therapies.

# Methods

**Reagents and tools table**

| Reagent/Resource | Reference or Source | Identifier or Catalog Number |
|---|---|---|
| **Experimental Models** | | |
| *Caenorhabditis elegans* strains | This study | Table EV3 |
| **Recombinant DNA** | | |
| pBN1 *mCh::his-58* | Rodenas et al, 2012 | |
| pBN20 *emr-1::mCh* | Morales-Martinez et al, 2015 | |
| pBN180 sgRNA | This study | |
| pBN207 *dpy-10* sgRNA | This study | |
| pBN209 *dam::emr-1* | Cabianca et al, 2019 | |
| pBN215 sgRNA(E + F) | This study | |
| pBN240 *emr-1* 3′ sgRNA | This study | |
| pBN296 *dam::baf-1* | This study | |
| pBN298 *dam::baf-1(G12T)* | This study | |
| pBN313 *g > f > p::baf-1* | Munoz-Jimenez et al, 2017 | |
| pBN485 *emr-1* 5′ sgRNA | This study | |
| pBN566 *FRT emr-1::mCh FRT* | This study | |
| pCFJ90 *myo-2p::mCh* | Frøkjær-Jensen et al, 2008 | |
| pCFJ104 *myo-3p::mCh* | Frøkjær-Jensen et al, 2008 | |
| pCFJ601 *eft-3p::Mos1* | Frøkjær-Jensen et al, 2012 | |
| pDD162 sgRNA | Dickinson et al, 2013 | |
| pMLS328 Cre | Schwartz and Jorgensen, 2016 | |
| #1286 eft-3p::Cas9 | Friedland et al, 2013 | |
| **Oligonucleotides and other sequence-based reagents** | | |
| Oligonucleotides and gBlocks | This study | Table EV4 |
| NEBNext® Multiplex Oligos for Illumina® (Index Primers Set 1) | New Englands BioLabs | E7335S |
| NEBNext® Multiplex Oligos for Illumina® (Index Primers Set 2) | New Englands BioLabs | E7500S |
| **Chemicals, Enzymes and other reagents** | | |
| Taq Polymerase | CABD Proteomics Facility | |
| Q5 polymerase | New Englands BioLabs | B9027S |

| Reagent/Resource | Reference or Source | Identifier or Catalog Number |
|---|---|---|
| Cas9 | CABD Proteomics Facility | |
| *BglII* | New Englands BioLabs | R0144S |
| *NheI* | New Englands BioLabs | R3131S |
| *Glycogen* | Roche | 10901393001 |
| *DpnI* | New Englands BioLabs | R0176S |
| *DpnII* | New Englands BioLabs | R0543S |
| T4 DNA ligase | Roche | 10799009001 |
| RNAse A | QIAGEN | 19101 |
| Taq DNA Polymerase | New Englands BioLabs | M0267S |
| Qubit | Invitrogen | Q32854 |
| PureLinkTM | Invitrogen miniprep kit | K2100-01 |
| QIAquick® PCR purification kit | QIAGEN | 28104 |
| NEBNext® Ultra DNA Library Prep Kit for Illumina® | New Englands BioLabs | E7370 |
| DNeasy Blood and Tissue Kit | QIAGEN | 69504 |
| SpeedBead Magnetic Carboxylate Modified Particles | GE Healthcare | 65152105050250 |
| 5-Fluoro-2'-deoxyuridine | Alfa Aesar | L16497 |
| tert-butylhydroperoxide | ThermoFisher Scientific | 458139 |
| **Software** | | |
| Rstudio v2022.12.0 | https://posit.co/download/rstudio-desktop/ | |
| damid.seq.R pipeline v0.1.3 | https://github.com/damidseq/RDamIDSeq | |
| damidseq_pipeline | https://owenjm.github.io/damidseq_pipeline/ | |
| polii.gene.call | https://github.com/owenjm/polii.gene.call | |
| WormBase Gene Set Enrichment Analysis | https://wormbase.org/tools/enrichment/tea/tea.cgi | |
| Genomics Viewer IGV Web App | https://igv.org/ | |
| jvenn Web App | https://jvenn.toulouse.inrae.fr/app/index.html | |

## Methods and protocols

### Nematode strains and transgenesis

Strains were maintained at 16 °C or 20 °C on Nematode Growth Medium (NGM) petri dishes seeded with *Escherichia coli* OP50 using standard *C. elegans* methods (Stiernagle, 2006). New strains were generated by MosSCI (Frøkjær-Jensen et al, 2012; Dobrzynska et al, 2016a), CRISPR/Cas9 (Arribere et al, 2014; Paix et al, 2015; Schwartz and Jorgensen, 2016) and crossing of existing strains. Descriptions and genotypes of all strains are listed in Table EV3. All strains are available from the authors upon request. The

N2 strain from the Caenorhabditis Genetics Center (CGC) was used as wild-type reference.

### Molecular cloning

Plasmids pBN296 *hsp16.41p > mCh::his-58>dam::baf-1* and pBN298 *hsp16.41p > mCh::his-58>dam::baf-1(G12T)* for DamID were constructed by amplifying *baf-1* and *baf-1(G12T)* with primers B925 and B926 (see primer sequences in Table EV4) using genomic DNA as template. The PCR product was digested with BglII and NheI and inserted into plasmid pBN209 *hsp16.41p > mCh::his-58>dam::emr-1* (Cabianca et al, 2019) digested with the same enzymes to replace *emr-1*.

Plasmid pBN566 *emr-1p::emr-1::mCh* with Frt sites in the first introns of *emr-1* and mCh was made by insertion of a synthetic gBlock B840 into plasmid pBN20 (Morales-Martinez et al, 2015), which contains 1568 bp upstream of the *emr-1* start codon, the *emr-1* CDS, mCh and 301 bp downstream of the *emr-1* stop codon.

Plasmid pBN180 to insert and express sgRNAs from the promoter of R07E5.16 (WBGene00014484) was derived from pDD162 (Dickinson et al, 2013) by whole-plasmid PCR with NEB Q5 polymerase and primers B633 and B634, deleting the Cas9 cassette. Plasmid pBN180 was further modified by whole-plasmid PCR with primers B780 and B781 to generate pBN215 that enables expression of sgRNA(E + F) (Ward, 2015). Plasmids pBN207, pBN240, and pBN485 to express sgRNAs corresponding to *dpy-10*, *emr-1* 3'-end, and *emr-1* 5'-end, respectively, were generated by whole-plasmid PCR with primer B635 and either primer B724 (*dpy-10*), B825 (*emr-1* 3'-end) or B838 (*emr-1* 5'-end) using either pBN180 (*dpy-10*) or pBN215 (*emr-1*) as PCR template. All DNA oligonucleotides were purchased from IDT.

### CRISPR/Cas9 genome editing

Strain BN1007+/mT1 [umnIs34] II; baf-1(bq24[G > F > P::baf-1[G12T]])/mT1 [dpy-10(e128)] III was generated by microinjection of targeting plasmid pBN313 (Munoz-Jimenez et al, 2017) at 65 ng/µl into the gonads of BN868 baf-1(bq19[G12T]) unc-119(ed3) III young adult hermaphrodites together with plasmids #1286 eft-3p::Cas9 ((Friedland et al, 2013); 25 ng/µl), pBN1 lmn-1p::mCh::his-58 ((Rodenas et al, 2012); 10 ng/µl), pCFJ90 myo-2p::mCh ((Frøkjær-Jensen et al, 2008); 2.5 ng/µl) and pCFJ104 myo-3p::mCh ((Frøkjær-Jensen et al, 2008); 5 ng/µl). Successful modification of the baf-1(G12T) locus was confirmed by detection of fluorescent fusion protein in the NE, absence of ectopic mCherry expression from the co-injection markers and Sanger sequencing. The unc-119(+) selection marker was next excised by microinjection with Cre expression plasmid pMLS328 (Schwartz and Jorgensen, 2016). Finally, uncoordinated hermaphrodites were outcrossed to N2 males to remove the unc-119(ed3) allele and balanced with mT1.

Strain BN1024 emr-1(bq34[emr-1::mCh]) I was generated by microinjection of repair template plasmid pBN566 at 50 ng/µl into the gonads of N2 young adult hermaphrodites together with plasmid #1286 eft-3p::Cas9 (50 ng/µl), sgRNA plasmids (pBN207, pBN240 and pBN485; 25 ng/µl each) and dpy-10 single-stranded repair template B725 (500 nM). Successful modification of the emr-1(G12T) locus was confirmed by detection of fluorescent fusion protein in the NE and Sanger sequencing followed by outcrossing twice to N2.

Strain PMX375 was generated by CRISPR-mediated gene editing, introducing the mutation c.34_35GG > AC into the

endogenous *baf-1* locus, and then six times outcrossed to N2, to eventually produce the strain BN808 *baf-1(bq19[G12T]) III*.

### Mos1-mediated single-copy integration

Single-copy insertion of tissue-specific DamID constructs into the *ttTi5605* locus on chrII was done by microinjection into the gonads of EG4322 young adult hermaphrodites. Plasmids pBN296 and pBN298 carrying *unc-119(+)* as selection marker were injected at 50 ng/µl together with pCFJ601 *eft-3p::Mos1* ((Frøkjær-Jensen et al, 2012), 50 ng/µl), pBN1 (10 ng/µl), pCFJ90 (2.5 ng/µl) and pCFJ104 (5 ng/µl). Integration of the constructs was confirmed by PCR and heat-inducible expression of mCh::HIS-58.

### Lifespan assay

Lifespan assays were performed as described (Haithcock et al, 2005). Animals were examined every 1–3 days and transferred to fresh plates every day for the first 5 days and then every 5–7 days. For experiments involving RNA interference (RNAi), plates contained 1 mM IPTG and 25 µg/ml ampicillin were seeded with 100 µl of *E. coli* HT115 bacterial culture the previous day. Control and *daf-16* RNAi proceeded from the Ahringer RNAi library (Kamath et al, 2003). Animals that crawled off the plate or displayed extruded internal organs or internal hatching were censored. Animals were considered dead when they failed to respond to gentle touch. Kaplan–Meir curves were prepared and analyzed with R (R version 3.6.3) and R Studio (Version 1.1.463; (Team, 2020)). All experiments were performed at least in triplicate and combined, except the results represented in Fig. EV1D. Statistical analysis was performed by using the R package survival that provides the chi-square statistic for a test of equality.

### Brood size assay

Three to five L4 worms per strain and per replica were individually picked on NGM plates seeded with *E. coli* OP50 and changed every 24 h until eggs laying ceased. Parental animal that crawled off the plate, dug into the agar, or suffered internal hatching precociously were censored. Progeny was counted after 2 days of development. All experiments were performed at least by triplicate. Statistical analysis was performed with R Studio using a two-tailed t-test.

### Nuclear morphology assay

Three to five hermaphrodites per genotype grown at 25 °C since L4 were observed at days 1, 6, and 8 after reaching L4 on each replica. Animals were anaesthetized with 10 mM levamisole and mounted on thin 2% agarose pads (de la Cruz Ruiz et al, 2022). A Nikon A1R microscope equipped with a 60× objective was used to acquire confocal micrographs of hypodermal nuclei every 1–2 µm from the surface of the worm to cover the entire depth of the hypodermis. Nuclear morphology was classified into four groups without revealing the genotype of the specimens to the researcher: class I) nuclei with uniform and smooth NE staining; class II) nuclei with enhanced intranuclear staining of NE components and mildly lobulated nuclei; class III) nuclei with highly irregular nuclear shape and accumulation of NE protein in bright foci concomitantly with reduced peripheral signal; class IV) nuclei similar to class III but with protrusions of the NE or ruptured NEs that release chromatin to the cytoplasm. All experiments were performed at least by triplicate. Statistical analysis was performed by using Fisher's exact test in R Studio.

### Quantification of fluorescence intensity

To quantify the fluorescence intensity of GFP::LMN-1, EMR-1::mCh, GFP::BAF-1, and GFP::BAF-1(G12T), images of hypodermal nuclei were acquired as above and fluorescence intensity was quantified with Fiji/ImageJ 2.1.0/1.53c (Schindelin et al, 2012) using the function Plot Profile of a manually drawn, 5 pixel wide straight line across the entire nucleus. NE values were obtained by averaging 3 consecutive values centered around the highest value on each side of the nucleus. Nucleoplasm intensity values were determined by averaging consecutive 5 values from the middle between the 2 NE maxima. Cytoplasm intensity values were determined by averaging 3 consecutive values 1 µm away from the NE on each side of the nucleus. Images of aged-matched N2 worms were used as background. Statistical analysis was performed with R Studio using a two-tailed t-test. Whereas quantification was performed on raw 12 bits images, identical adjustment of brightness and contrast was applied to all comparable panels within each figure panels.

### Time-lapse recordings

Embryos were obtained by cutting open gravid hermaphrodites using two 21-gauge needles, mounted on a coverslip in 3 µl of M9 buffer. The coverslip was placed on a 2% agarose pad. For high temporal resolution videos, 1 cell-stage embryos were recorded with a Yokogawa CSU-X1 spinning disc confocal scanning unit mounted on a Olympus IX-81 microscope equipped with 491 nm and 561 nm lasers and a 100× objective. Acquisition parameters were controlled by MetaMorph software (v7.8.13.0). Z stack series consisting of 17 slices separated by 0.3 µm were acquired every 12.5–20.4 s followed by maximum projection for each time point. For recordings of the entire embryonic development, coverslips were sealed with VALAP to avoid desiccation and 5 confocal sections separated by 2 µm were acquired every 10 min on a Nikon A1R microscope equipped with a 60× objective, Nikon Perfect Focusing System and 488 nm and 561 nm lasers. Single confocal sections were used for mounting figures. Room temperature was 20–23 °C. Images were processed using FIJI software, always applying identical settings to control and mutant recordings.

### RNA purification

Eggs were obtained by sodium hypochlorite treatment and incubated in M9 containing 0.01% Tween-20 (M9-T) overnight at 20 °C without food. For each strain, three 60 mm NGM plates with a lawn of OP50 *E. coli* were seeded each with approximately 500 synchronized L1 larvae and incubated at 20 °C for 48 h until. L4 larvae were washed off the plates with M9-T, collected in 15 ml tubes and washed three times in M9-T. Finally, worms were pelleted in 1.5 ml tubes. The supernatant was aspired to leave ~50 µl and 500 µl Trizol was added. Tubes were snap-frozen in liquid nitrogen and stored at −80 °C.

Samples were lysed in 5 freeze/thaw cycles (1 min in liquid nitrogen; 3 min in 37 °C shaker) and left for 3 min at RT. Then, 100 µl chloroform was added, tubes were shaken vigorously for 15 s and left for 3 min at RT. After centrifugation at 12,000 × *g*, 4 °C for 15 min, the aqueous phases were transferred to new tubes containing 0.5 µl 20 mg/ml glycogen. RNA was precipitated by adding 250 µl isopropanol and centrifugation at 12,000 × *g*, 4 °C for 10 min. Supernatants were removed and pellets were washed in 500 µl ethanol 75%. After centrifugation and removal of the

supernatants, the pellets were left to dry for 5 min and resuspended in DEPC-treated water. Residual DNA was digested by treatment with DNase I (NEB) in a total volume of 50 µl at 37 °C for 10 min. DNase I was inactivated by addition of 0.5 µl 0.5 M EDTA. RNA was purified with NEB Monach spin columns, eluted in 20 µl DEPC-treated water and evaluated on a Nanodrop.

### First-strand cDNA synthesis and quantitative PCR

cDNA was prepared from 600 ng total RNA in a final volume of 40 µl using NZY First-Strand cDNA Synthesis kit (NZYtech). Mixes were incubated at 25 °C for 10 min and 55 °C during 30 min. Reactions were inactivated by heating for 5 min at 85 °C. RNA was eliminated by adding 1 µl of NZY RNase H during 30 min at 37 °C. Quantitative PCRs were performed using NZYSupreme qPCR Green Master Mix (2x) (see primer sequences in Table EV4). 2 µl of cDNA were added in a final volume of 10 µl per reaction. Gene expression was evaluated in three independent biological replicas, with three technical replicas each. The PCR program consisted of 2 min at 95 °C followed by 40 cycles of two steps, 5 s at 95 °C and 30 s at 60 °C. Melting curves were determined by incubation at 65 °C for 5 s and at 95 °C for 50 s.

### Resistance to ultraviolet irradiation

For every replica, 12–15 L4 hermaphrodites were isolated and grown at 20 °C. Next day, five of these worms were transferred to a new dish an irradiated in a Hoefer UVC 500 UV Crosslinker, with a dose of 25 J/m². In parallel, another five worms were isolated but were not exposed to UV. Both, irradiated and non-irradiated worms were incubated for 2 h at 20 °C. Then, after removing the adults, eggs were counted, and plates were incubated at 20 °C. The day after, larvae and non-hatched eggs were quantified. Statistical analysis was performed with R Studio using a two-tailed paired t-test.

### Heat stress assay

L4 worms were picked the day before the assay and maintained overnight at 20 °C. The day of the assay, thermotolerance resistance was evaluated by transferring 20 young adults to an NGM plate without bacteria and incubated in a water bath at 35 °C. Viability was scored every hour after 2 h and death was determined by the lack of movement and/or pharyngeal pumping after gentle prodding. The experiment was performed in triplicate and the combined data were analyzed as described for lifespan above.

### Oxidative stress assay

The animals were synchronized as L1 larvae by egg-prepping with hypochlorite (Stiernagle, 2006) and hatching in M9. L1 larvae were then seeded onto 6 cm NGM plates containing 1 mM IPTG and 50 µg/ml carbenicillin seeded with E. coli HT115 (transformed with the L4440 plasmid). Animals were grown at 15 °C until the late L4 stage, followed by the addition of 5-Fluoro-2'-deoxyuridine (50 µM final concentration) and shifted to 20 °C. At day 2 of the adulthood, animals were washed off the plates with M9 and transferred to plates containing 6 mM tert-butyl hydroperoxide (tBOOH). Seeding density was ~50 animals per plate. Plates were then incubated at 20 °C and the animals' survival was recorded and analyzed by a fully automated lifespan machine as previously described (Stroustrup et al, 2013). For each condition, a minimum

of 150 worms distributed over at least 4 separate plates was scored. The combined data were analyzed as described for lifespan above.

### Bacteria culture for DamID plates

E. coli GM119 dam- was grown by pre-inoculating 1–2 colonies from a fresh LB/agar plate in liquid LB without antibiotics. The pre-inoculum was grown overnight at 37 °C before transferring 1 ml of culture to a suitable volume of LB (depending on the numbers of plates to be seeded, i.e., for 10 plates with a diameter of 10 cm prepare at least 10 ml culture) and incubated for 6 h at 37 °C. Next, 700 µl of bacteria culture was added to each 10 cm NGM plate, spread and incubated overnight at 37 °C.

### Nematode culture for DamID

Worms were grown asynchronously at 20 °C on NGM plates seeded with E. coli GM119 for at least two generations before egg-prepping with hypochlorite. Hatched L1s were counted the day after and 1000 L1s were aliquoted onto 10 cm NGM plates seeded with E. coli GM119 as described above. Worms were grown at 20 °C for 48 h before collecting 4000 L4 stage nematodes per strain and washed 8 times with 15 ml of M9 with 0.01% Tween-20. Worms were transferred to 1.5 ml tubes with M9 with 0.01% Tween-20 using Pasteur pipettes followed by two additional washes. Supernatants were aspirated and aliquots of 30 µl were snap-frozen in liquid nitrogen and maintained at −80 °C until DNA extraction.

### DamID amplification, library preparation, and sequencing

To purify genomic DNA, samples were lysed in 5 freeze/thaw cycles (1 min in liquid nitrogen; 3 min in 37 C shaker) and processed with DNeasy Blood and Tissue Kit (QIAGEN #69504). DamID was performed on 200–400 ng genomic DNA (gDNA; 200 ng was used for RNA polymerase DamID [RAPID]; 400 ng was used for BAF-1 and BAF-1(G12T) DamID) extracted from 4000 L4 animals cultured and collected as indicated above. Two (RAPID) or three (BAF-1 and BAF-1(G12T) DamID) replicates were performed as described (de la Cruz Ruiz et al, 2022). For RAPID, a pool of 3 AdR primers (B1436 AdR4N, B1437 AdR5N, B1438 AdR6N) and 14 PCR cycles were used. For BAF-1 and BAF-1(G12T) DamID we used B916 AdR and 22 PCR cycles. Successful amplification of Dam-methylated genomic DNA yielded a smear of ~400–1500 bp fragments when analyzed on Agilent Bioanalyzer.

Either 40 ng (RAPID) or 400 ng (BAF-1 and BAF-1(G12T) DamID) of PCR amplified fragments were used for library preparation as described (de la Cruz Ruiz et al, 2022), using E7370 NEBNext® Ultra™ DNA Library Prep Kit for Illumina. Library amplicons' size distribution was checked to be maintained as for PCR fragments indicated above followed by pooling, repurification and sequencing on a Illumina NextSeq500 platform at EMBL GeneCore.

### Bioinformatic analysis

FASTQ files from the NextSeq500 platform were processed in R Studio using the pipelines RDamIDSeq (https://github.com/damidseq/RDamIDSeq) and RGeneDamIDSeq (https://github.com/damidseq/-RGeneDamIDSeq-) (Sharma et al, 2016) for mapping of reads to bins of fixed size (e.g., 2 kb, 10 kb, or 100 kb) and to genes, respectively. The pipelines include a quality check where only reads that contain the DamID adapter ("CGCGGCCGAG") and map to

GATC sites in the *C. elegans* genome are retained for further analysis. The number of reads fulfilling these criteria ranged from 0.5 to 27 million (see Table EV5). Both pipelines also provide mapped read numbers per individual GATC fragment. For mapping to genes, the script was modified to include 500 bp distal to transcriptional start and termination sites. UCSC genome version ce11 (BSgenome.Cele-gans.UCSC.ce11) was used for alignment.

For BAF-1 and BAF-1(G12T) DamID the DESeq2 package (Love et al, 2014) was used to identify bins with significantly more reads with either Dam::BAF-1 or Dam::BAF-1(G12T) compared to GFP::Dam across the 3 replicas. A false discovery rate (FDR) of 0.05 was used as threshold. Next, for each sample a pseudocount of 1 was added to each bin and the relative number of reads per bin was calculated followed by averaging across the replicas. The ratio of Dam::BAF-1 or Dam::BAF-1(G12T) to GFP::Dam was calculated for each bin and log2-transformed. Finally, normalization was done by subtracting the genome-wide average log2 ratio from the log2 ratio of each bin (Brueckner et al, 2020).

For RAPID, normalization of Dam::RPB-6 reads to GFP::Dam reads was performed using the damidseq_pipeline package (https://owenjm.github.io/damidseq_pipeline/) (Marshall and Brand, 2015) on the bam files generated by the RDamIDSeq pipeline. Gene occupancy was computed as log2(Dam::RPB-6/GFP::Dam) with polii.gene.call (https://github.com/owenjm/polii.gene.call) using bedgraph output files from the damidseq_pipeline and WBCel235 gene annotation. A FDR of 0.05 was used as threshold to call transcribed genes.

For comparison of chromosome centers versus heterochromatin-rich distal arms, we used the following border coordinates: ChrI 3745632 and 10809938, ChrII 4708341 and 11877168, ChrIII 3508994 and 9947268, ChrIV 7317812 and 12176625, ChrV 8125434 and 13849337, and ChrX 41919362 (Gonzalez-Aguilera et al, 2014). Pairwise Wilcoxon rank test values were calculated in R.

Genome browser views and Venn diagrams were generated with the University of California Santa Cruz Genomics Institute Genome Browser (http://genome.ucsc.edu; (Kent et al, 2002)) and jvenn (Web App) (Bardou et al, 2014), respectively. The WormBase Enrichment Analysis Suite (http://www.wormbase.org/tools/enrichment/tea/tea.cgi; (Angeles-Albores et al, 2018)) was used for tissue enrichment and gene ontology analysis.

## Data availability

The datasets produced in this study are available in the following databases: RAPID data: Gene Expression Omnibus GSE261819. BAF-1 and BAF-1(G12T) DamID data: Gene Expression Omnibus GSE261820.

The source data of this paper are collected in the following database record: biostudies:S-SCDT-10_1038-S44318-024-00261-8.

## Peer review information

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

## Acknowledgements

We are grateful to Marta Rojas for assistance in genetic crosses, Laura Tomás from the CABD Proteomics and Biochemistry Facility for recombinant Cas9,

Katherina García from the CABD Imaging Facility and Ildefonso Cases from the CABD Bioinformatics Facility for technical assistance, Vladimir Benes from EMBL GeneCore for advice on sequencing of DamID libraries and Delphine Larrieu for insightful comments. We also acknowledge the Caenorhabditis Genetics Center (CGC), which is funded by the National Institutes of Health (NIH) Office of Research Infrastructure Programs (P40 OD010440) and WormBase. This project was funded by the Spanish Ministerio de Ciencia, Innovación y Universidades (MCIU), Agencia Estatal de Investigación, the European Union, the European Regional Development Fund (PID2019-105069GB-I00, CEX2020-001088-M, and PID2022-137162NB-I00; doi:10.13039/501100011033) and the Spanish National Research Council, (2023AEP058). RRB was supported by contract BES-2017-080216 from the Spanish MCIU and Scientific Exchange Grant 9276 from EMBO. JDW was supported by the National Science Foundation (NSF) Division of Molecular and Cellular Biosciences (CAREER award 1942922). CGR was supported by the Swedish Research Council (VR) (VR2023-04383), the Swedish Cancer Society (Cancerfonden) (23 2994 Pj), the Novo Nordisk Foundation (NNF22OC0078353). AK was supported by a Karolinska Institutet Doctoral Education Grant (KID).

## Author contributions

**Raquel Romero-Bueno**: Conceptualization; Resources; Data curation; Formal analysis; Validation; Investigation; Visualization; Methodology; Writing—original draft; Writing—review and editing. **Adrián Fragoso-Luna**: Conceptualization; Resources; Data curation; Formal analysis; Validation; Investigation; Visualization; Methodology; Writing—review and editing. **Cristina Ayuso**: Resources; Data curation; Investigation; Methodology. **Nina Mellmann**: Formal analysis; Investigation; Methodology. **Alan Kavsek**: Formal analysis; Investigation; Methodology. **Christian G Riedel**: Formal analysis; Funding acquisition; Investigation; Methodology. **Jordan D Ward**: Formal analysis; Funding acquisition; Investigation; Methodology; Project administration. **Peter Askjaer**: Conceptualization; Resources; Data curation; Formal analysis; Supervision; Funding acquisition; Validation; Investigation; Visualization; Methodology; Writing—original draft; Project administration; Writing—review and editing.

Source data underlying figure panels in this paper may have individual authorship assigned. Where available, figure panel/source data authorship is listed in the following database record: biostudies:S-SCDT-10_1038-S44318-024-00261-8.

## Disclosure and competing interests statement

The authors declare no competing interests.

# Expanded View Figures

**Figure EV1.  *baf-1(G12T)* mutants have normal viability and lifespan at 20 °C.**                                ▶

(**A**) The predicted 3-dimensional structure of *C. elegans* BAF-1 (rainbow-colored; Uniprot Q03565) was superimposed onto the structure of human BAF (magenta; Uniprot O75531) using iCn3D Structure Viewer (Wang et al, 2022) (https://www.ncbi.nlm.nih.gov/Structure/icn3d/). Residue 12 is highlighted in green. (**B, C**) The egg laying rate (**B**) and embryonic viability (**C**) were determined for hermaphrodites grown at 20 °C, 25 °C or shifted from 20 °C to 25 °C when reaching adulthood. (**D**) Lifespan assay of wild-type and *baf-1(G12T)* hermaphrodites at 20 °C. The number of animals analyzed is indicated in brackets. *P* values from two-sided t-test (**B, C**) and Chi-square test (**D**) are indicated; *p* values ≥ 0.05 are considered not significant (n.s.).

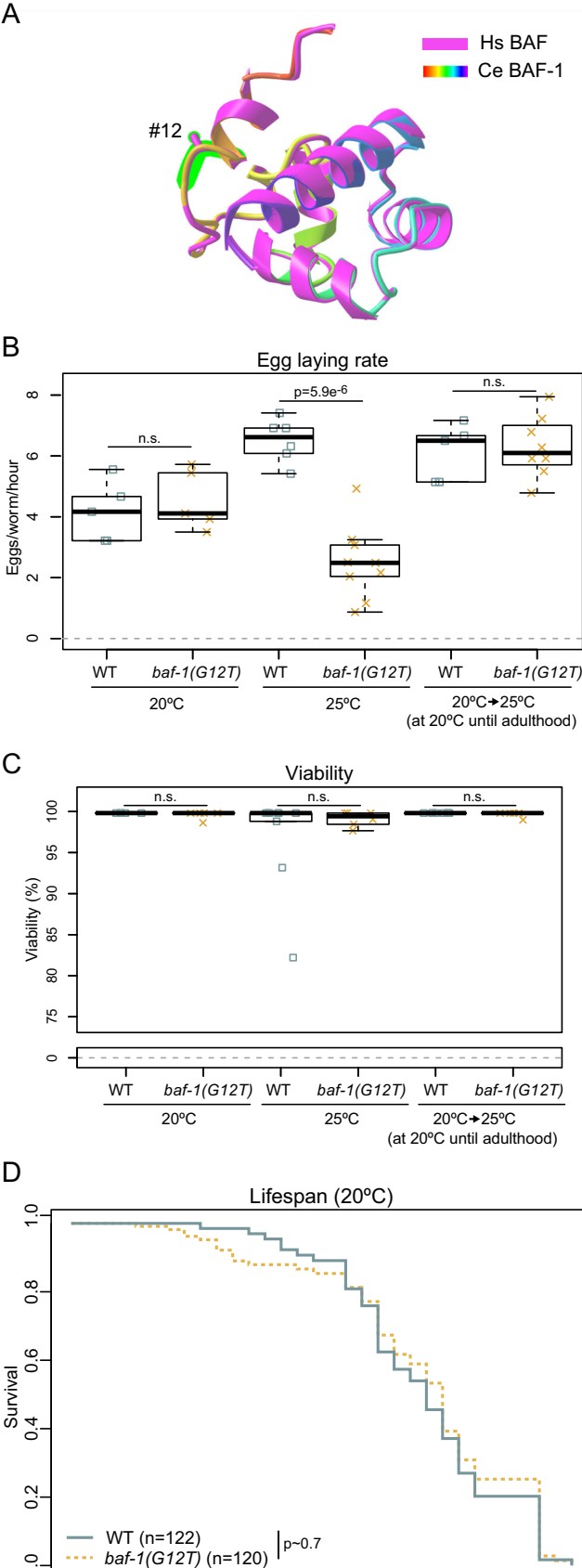

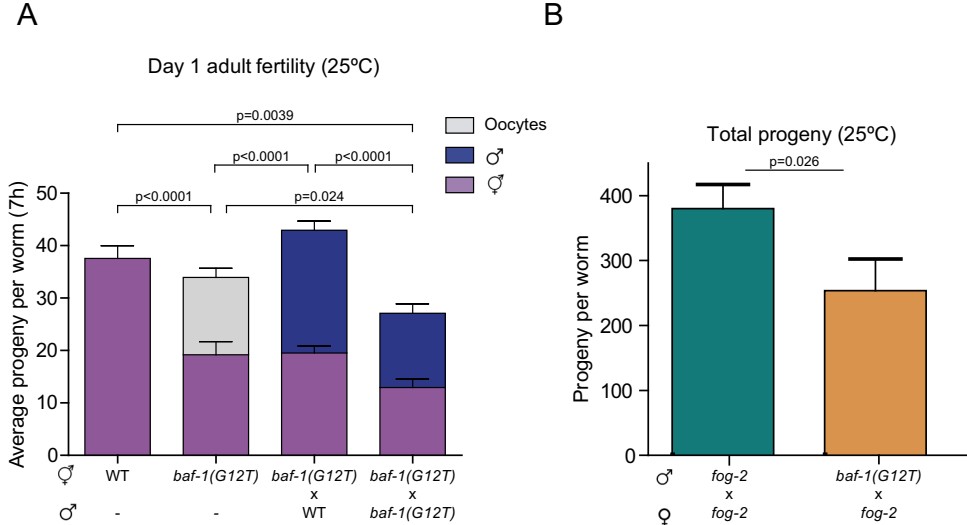

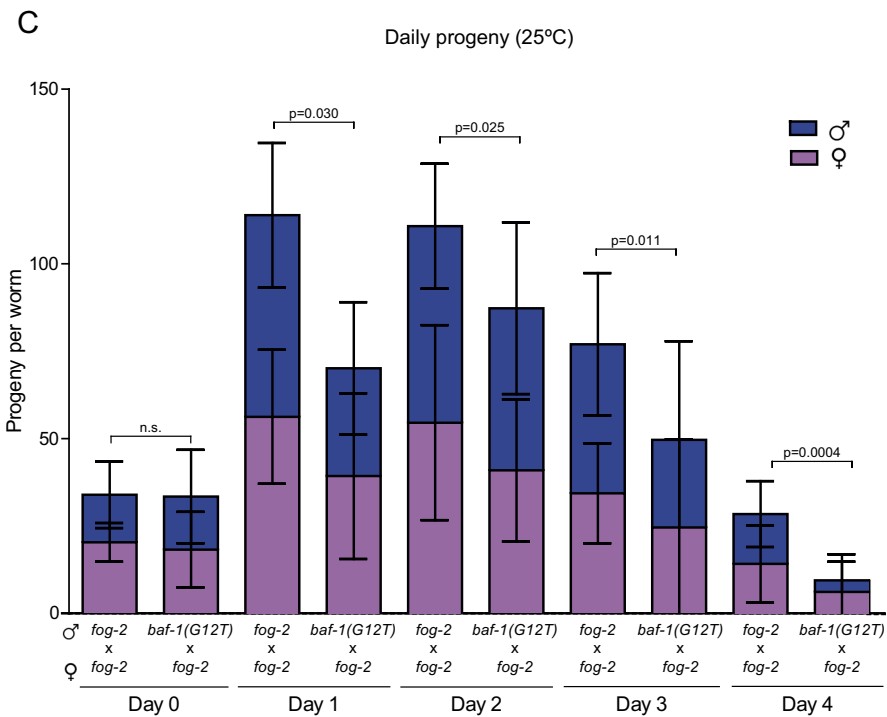

**Figure EV2.  Sperm defect caused by *baf-1(G12T)* mutation.**

(A) Progeny produced at 25 °C by day 1 wild-type and *baf-1(G12T)* adults either unmated or mated with wild-type or *baf-1(G12T)* males as indicated. (B, C) Total (B) and daily (C) brood size produced at 25 °C by feminized *fog-2* animals mated with either *fog-2* or *baf-1(G12T)* males. Error bars represent standard deviation. *P* values from two-sided t-test are indicated; *p* values ≥ 0.05 are considered not significant (n.s.).

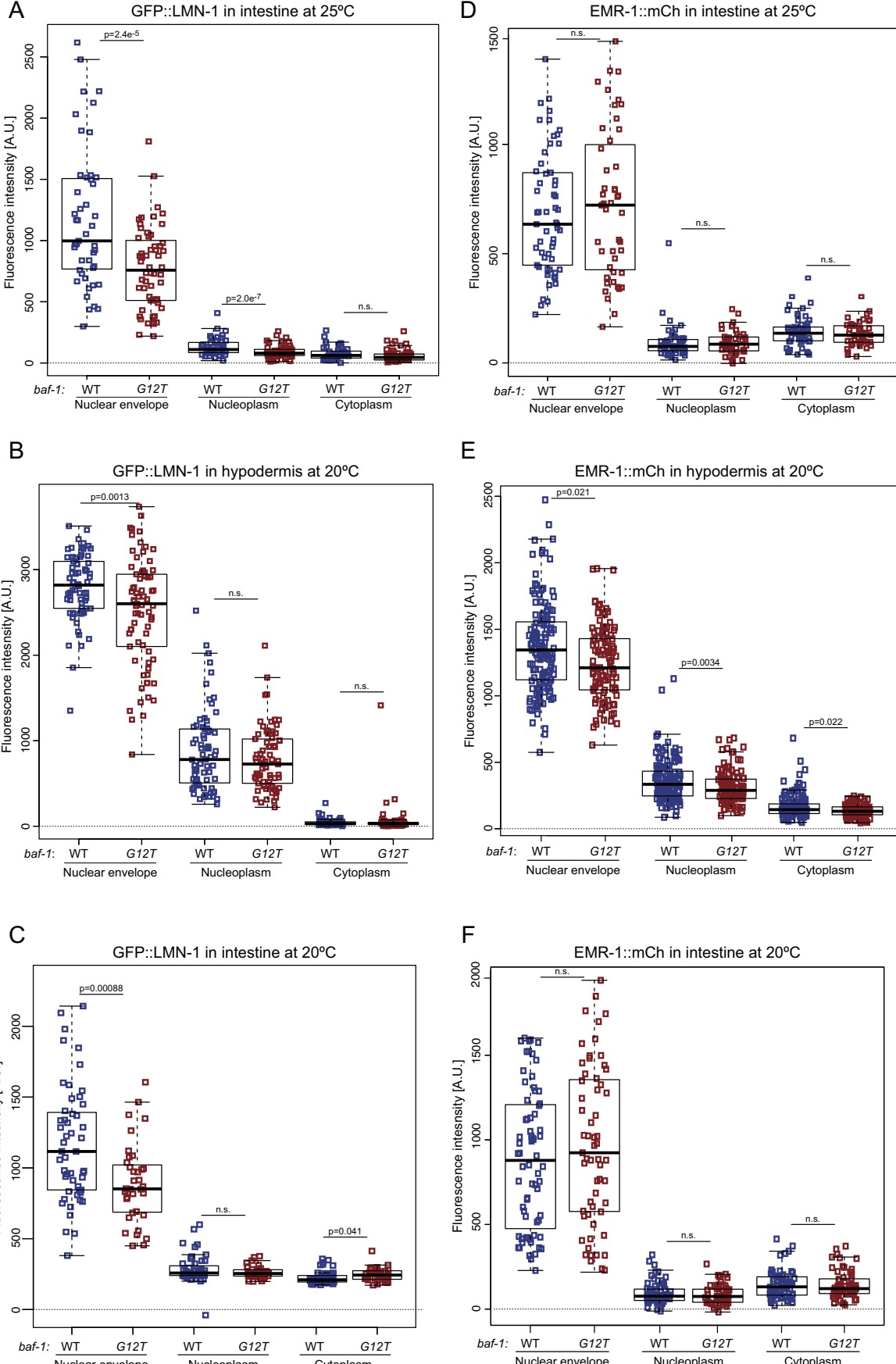

◀ **Figure EV3. Nuclear envelope accumulation of LMN-1 is decreased in *baf-1(G12T)* mutants.**

Quantification of GFP::LMN-1 (**A–C**) and EMR-1::mCh (**D–F**) signal in the NE, nucleoplasm and cytoplasm of intestinal and hypodermal cells at 20 °C and 25 °C as indicated. Data from 3 independent experiments; at least 15 animals were analyzed for each strain. *P* values from two-sided t-tests are indicated; *p* values ≥ 0.05 are considered not significant (n.s.).

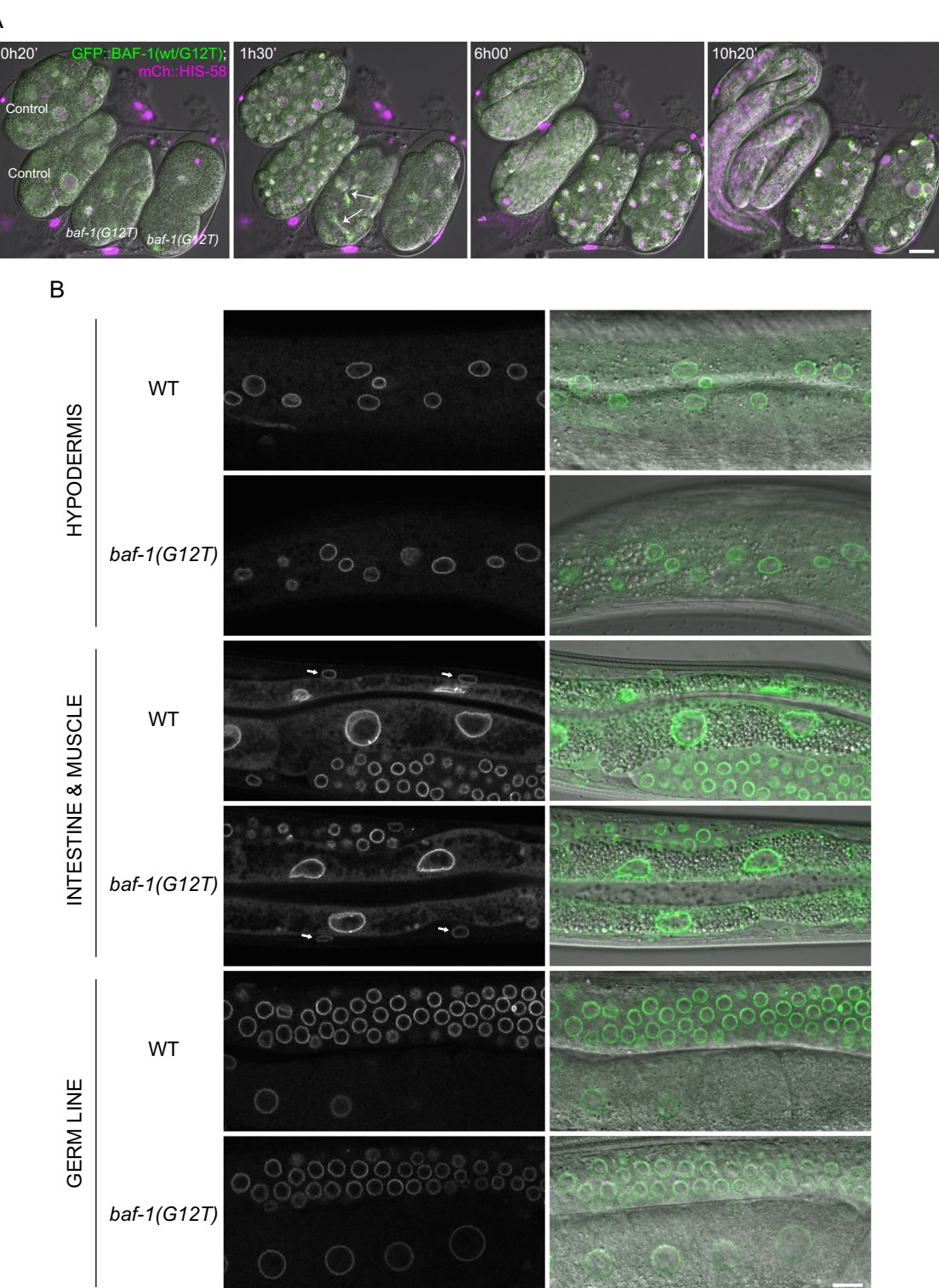

**Figure EV4. *baf-1(G12T)* worms are hypersensitive to GFP tagging.**

(**A**) Selected time points from confocal time-lapse recording of 2 GFP::BAF-1 and 2 GFP::BAF-1(G12T) endogenously tagged embryos co-expressing mCh::HIS-58 (magenta). Arrows indicate failures in chromosome segregation. (**B**) Confocal micrographs of hermaphrodites expressing endogenously tagged GFP::BAF-1 or GFP::BAF-1(G12T) as indicated. Scale bars represent 10 μm.

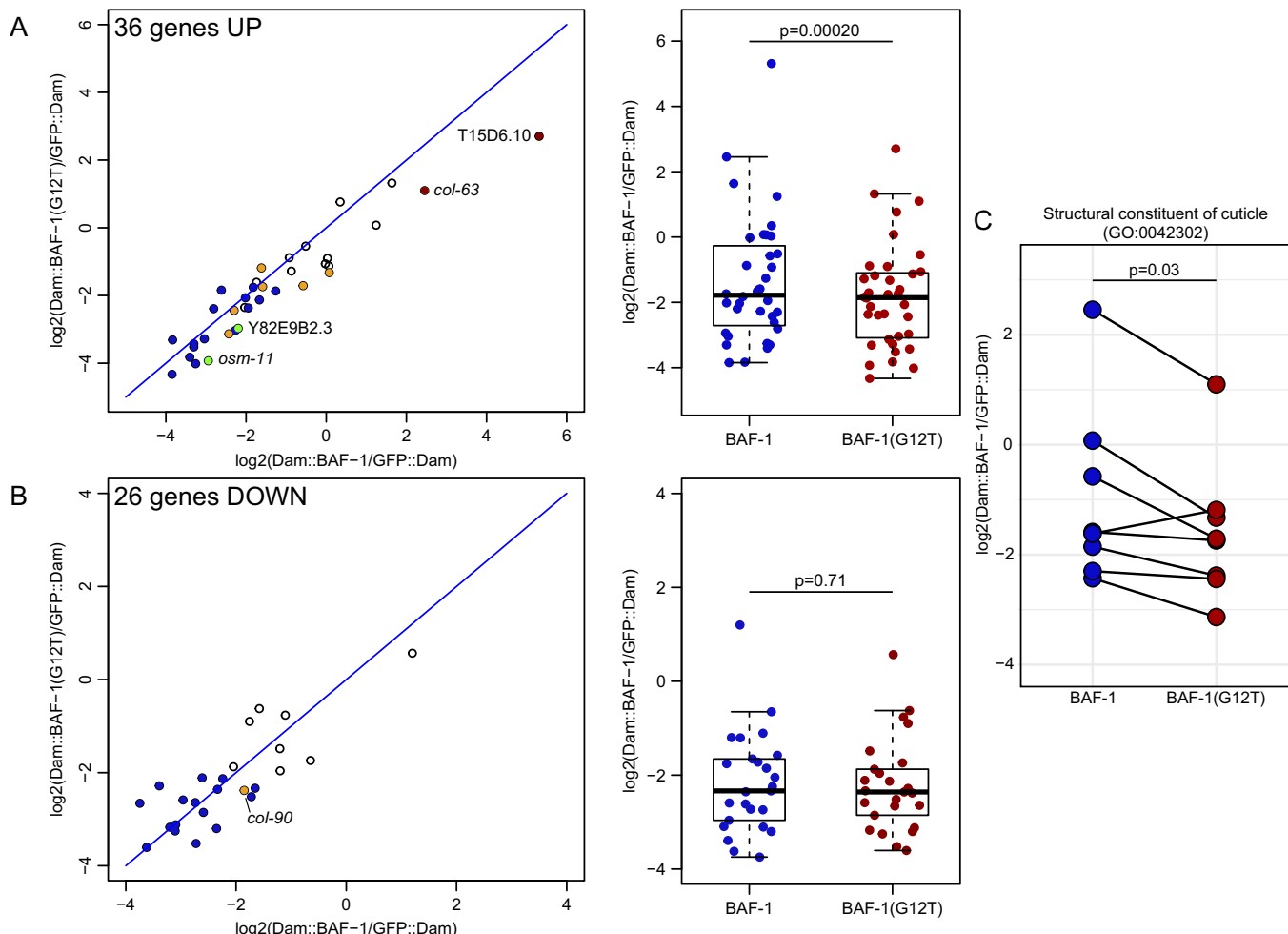

**Figure EV5.  Differentially expressed genes encoding cuticle components bind distinctively to BAF-1 and BAF-1(G12T).**

(**A, B**) Genes with increased (**A**) or decreased (**B**) expression in hypodermis of *baf-1(G12T)* mutants were analyzed for association to wild-type BAF-1 (x axis) and BAF-1(G12T) in scatter plots (left; BAF-1 association on x axis; BAF-1(G12T) association on y axis) and boxplots (right). Genes with significantly higher association to Dam::BAF-1 or Dam::BAF-1(G12T) than to GFP::Dam are indicated in red in scatter plots. Genes indicated in orange (and *col-63* indicated in red) encode structural constituents of the cuticle. Genes plotted in green are also deregulated in intestine (see Fig. 6). (**C**) The association of differentially expressed genes encoding structural constituents of the cuticle to BAF-1 and BAF-1(G12T) in hypodermis. *P* values from two-sided paired t-tests, excluding outliers in (**A, B**), are indicated.

