## [Peer Review File · The EMBO Journal]

A human progeria-associated BAF-1 mutation modulates gene expression and accelerates aging in *C. elegans*

Raquel Romero-Bueno, Adrian Fragoso-Luna, Cristina Ayuso, Nina Mellmann, Alan Kavsek, Christian Riedel, Jordan Ward, and Peter Askjaer

Corresponding author(s): Peter Askjaer (pask@upo.es)

Review Timeline:

Transfer from Review Commons:	11th Jul 24
Editorial Decision:	21st Aug 24
Revision Received:	9th Sep 24
Accepted:	17th Sep 24

Review
COMMONS

Editor: Kelly Anderson

Transaction Report: This manuscript was transferred to The EMBO JOURNAL following peer review at Review Commons.

Review #1

1. Evidence, reproducibility and clarity:

Evidence, reproducibility and clarity (Required)

****Summary:****

Specific mutations in nuclear lamina proteins such as lamin A or BAF1 can cause premature aging syndromes (Hutchinson Gilford progeria or Nestor Guillermo syndrome), which show age related deterioration of nuclear morphology as a hallmark and affected individuals have a severely shortened life span. In this study Romero-Bueno et al established an animal model system for the Nestor Guillermo syndrome, by generating *C. elegans* strains harboring homozygous *baf-1::G12A* mutations. They nicely recapitulate the expected cellular and organismal phenotypes: decreased life spans of the mutant animals and faster nuclear deterioration. In addition, the authors find reduced fertility in the mutant when BAF-1 was combined with a GFP tag and synthetic lethality when the *baf-1::G12T* mutant is introduced into strain carrying an epitope tagged Lamin allele. At the organismal level the authors report increased resistance to oxidative stress, but reduced thermotolerance and decreased UV resistance. By conducting tissue specific DamID together with tissue specific RNA polymerase DamID, the authors find that in the *baf-1::G12T* mutant the overall chromatin association of chromosome arms remains largely unchanged, however a few individual loci either lost or gained BAF-1 association. The authors report that loss of BAF-1 association with chromatin correlates with gene expression in some instances, however there was no strict uniform correlation. This is not surprising, because the changes in expression could be a secondary consequence of a BAF-1-mediated change of another locus or a consequence of altered lamin nuclear envelope association. The global pattern of BAF-1 and BAF-1 G12T binding to chromatin was very similar in both genotypes. The authors find unaltered localization of BAF-1 G12T protein at the nuclear envelope, in contrast to reduced levels of lamin and emerin. Interestingly, BAF-1 is found on sperm, in contrast to the absence of other lamina proteins, like LMN-1 or Emerin.

****Major comments:****

1. In the *baf-1* G12T mutants the authors find reduced levels of lamin in hypodermal nuclei. It would be good to also examine the dynamics of lamin in the second tissue that was subjected to DamID (intestinal cells).
2. The authors make a statement that EMR-1 expression was reduced in the *baf-1* G12T mutant, but do not comment on LMN-1 expression. Can a statement on this be made by RT PCR?
3. The authors find few alterations in gene regulation of the loci which have different occupancy WT BAF-1 versus BAF-1 G12T. It was surprising to see the DamID and RNA polymerase DamID experiments be done with worms grown at 20°C, because the more penetrant phenotypes at the organismal level were observed at 25°C. Could this be the reason for the little change of chromatin occupancy of BAF-1 and BAF-1 G12T or few changes in gene expression? Would it make sense to examine the expression of some selected BAF-1 bound loci by single molecule Fish at 25°C and compare expression wt versus *baf-1* G12T?

****Minor comments:****

The finding that BAF-1 nuclear envelope localization remains unchanged in the mutant stems from detection of the inserted GFP epitope. Given that the tag has an influence on the BAF-1 G12T mutant

viability, this statement should be phrased with more care. The tag could influence the turnover of the protein for example.

Maybe Western blots comparing the signal of WT BAF-1 worms and BAF-1 G12T mutant worms would be instructive to compare the levels of the protein (at 20°C and at 25°C, day 1 adults and day 8 adults)

Line 105: typo: remove "s"

Line 154: A conclusion is missing for the fog-2 experiment

would it make sense to discuss a possible influence of altered lamin binding to the nuclear envelope in the mutant in the context of the gene expression results?

****Referees cross-commenting****

I also see this as a valuable study.

I regretted a bit that the analysis was not done at the higher temperature when the authors saw the most prominent phenotypes--but I suppose the analysis is very expensive and time consuming.

2. Significance:

Significance (Required)

Since a long time, it has remained a matter of debate whether the progeria T to G transition in BAF-1 reduces binding of BAF-1 to lamin or whether the mutation affects the binding of BAF-1 to chromatin and thereby alters chromatin organization. Conflicting results emerged from the studies of BAF1 mutants in tissue culture cells. For this reason, this study--conducted in the context of a whole animal--is very important: it allowed the author to do their experiments with cells with unaltered ploidy, expression from the endogenous promoters and in the context of defined tissues. A second conflicting finding concerned the localization of BAF-1 G12 to T mutant protein at the nuclear envelope: some labs find it reduced at the nuclear envelope, others find unchanged amounts at the nuclear envelope. With this work the authors contributed novel and interesting findings to those ongoing discussions, they found both altered affinity of BAF-1 with chromatin (not on a global scale, but on a local scale) and reduced affinity to lamin.

Furthermore, this is one of the first studies mapping BAF-1 association with individual gene loci in a specific tissue and the authors showed that in a given tissue BAF-1 tends to be associated with not expressed genes.

In a nutshell, the authors have established a convincing accessible model system for studying aging, ready for consecutive testing interventions to reduce the pace of premature aging.

Strengths: convincing presentation of a novel genetic model system to study progeria, first study where BAF-1 bound loci were shown from the analysis of a tissue (there is a correlation of BAF-1 bound loci, which are not expressed in the examined tissue), introduction of an easy-to-handle model to search for compounds suitable for clinical intervention for progeria patients or anti-aging drugs. This study adds some clarity to conflicting views in the field: the NGS mutation in BAF-1 both reduces the amount of lamin at the nuclear periphery and affects the affinity of BAF-1 to chromatin.

***Limitations:** the observed transcriptional changes in the mutant can be either a direct consequence of BAF-1 chromatin association or a consequence of an altered lamina since lamin is less stable at the nuclear envelope. The transcriptomic analysis was not conducted at the temperature at which penetrant phenotypes at the organismal level were observed.

Advance: Previous studies presented conflicting results about the nuclear envelope localization of BAF-1 G12T protein: this study clearly shows that the localization of the protein remains unaltered. This study also clearly demonstrates that there is less lamin at the nuclear envelope in the mutant, lending support to the in vitro findings that the mutant is compromised in Lamin binding.

Audience: The study will be of interest to anyone who studies the nuclear lamina, the nuclear envelope, progeria, aging and stress response of an organism. Beyond this a convincing powerful novel genetic system is being presented to study progeria, which is of interest to clinicians. It is also of interest for translational research, because the system can be used to screen for compounds, which could be useful for therapeutic intervention for progeria patients or for the identification of compounds that combat aging in general.

Some of the presented synthetic effects with tagged strains open the opportunity to conduct genetic suppressor screens, which would be a wonderful entry point to collect more mechanistic insights in the phenomena of aging and stress response. This genetic system is an awesome starting point for further studies and advancements of elucidating the molecular mechanism of progeria by genetic screens.

Expertise: I am a *C. elegans* geneticist and appreciate that all the conflicting results from tissue culture studies can now be compared to an analysis in a more physiological setting and the context of a real tissue and a living animal. I am not really competent to judge the sensitivity of RAPID assays.

3. How much time do you estimate the authors will need to complete the suggested revisions:

Estimated time to Complete Revisions (Required)

(Decision Recommendation)

Between 1 and 3 months

Yes

Review #2

1. Evidence, reproducibility and clarity:

Evidence, reproducibility and clarity (Required)

Romero-Bueno et al. have generated a *C.elegans* model of Nestor-Guillermo Progeria Syndrome (NGPS) caused by a point mutation in the BAF1 gene and they characterize the worm model. They find reduced

fertility, reduced longevity and earlier aging symptoms in mutant animals compared to wild type animals. Looking at the molecular level, the authors find reduced accumulation of lamin A and emerin at the nuclear periphery in cells from mutant animals. They also find mitotic chromosome segregation defects. Using tissue-specific DamID, they show altered binding patterns of the mutant protein to genome regions and they identify some gene groups which show both changes in gene expression and BAF1 binding. Finally, they show that BAF1 mutants are more resistant to oxidative stress than wild type animals.

The value of this work is two-fold: First, it is a very robust characterization of NGPS worms. Second, this will be a very useful model for the study of NGPS. Overall, the study is well-designed, technically strong, and the results are carefully and thoughtfully interpreted, which is nicely exemplified by the discussion of the relatively small number of genes which are differentially bound by BAF1 and are also differentially expressed and the authors do a good job of not overinterpreting the data, but simply state them. The results are convincing and informative.

My only minor point that may make this paper marginally better is that it would be nice to have a paragraph in the Discussion elaborating on the potential and the limitations of using the worm model to understand human NGPS, for example, humans have multiple lamin proteins etc.

****Referees cross-commenting****

I am glad to see that there is strong agreement that this is a valuable study.

2. Significance:

Significance (Required)

The study is significant as it introduces a new animal model system to study an ultra-rare disease. The presented results are robust and convincing. This is the first worm model for this disease and it will be of interest to those studying laminopathies. The worm model is expected to reflect some of the human disease phenotypes but not all and a discussion of the potential and the limitation of the worm model to study NGPS would be welcomed.

3. How much time do you estimate the authors will need to complete the suggested revisions:

Estimated time to Complete Revisions (Required)

(Decision Recommendation)

Less than 1 month

4. Review Commons values the work of reviewers and encourages them to get credit for their work. Select 'Yes' below to register your reviewing activity at Web of Science Reviewer Recognition Service (formerly Publons); note that the content of your review will not be visible on Web of Science.

No

Review #3

1. Evidence, reproducibility and clarity:

Evidence, reproducibility and clarity (Required)

The mechanisms of rare human progeria syndromes caused by mutations in nuclear lamina proteins (lamins or BANF1) are still poorly understood, mainly because these proteins are complicated: they interact and are structurally essential for mitosis and nuclear assembly; hence, disrupting either protein can (and often does) disrupt the other. Lamins and BANF1 also have multiple interwoven roles with other partners involved in 3D genome organization, chromatin regulation, and tissue-specific gene regulation during interphase. To focus on Nestor-Guillermo progeria syndrome (NGPS), caused by the homozygous A12T missense mutation in human BANF1, the authors inserted the corresponding G12T mutation in *C. elegans* baf-1. They tested potential phenotypes at multiple levels (molecular, transcriptional, cellular and organismal) extensively and rigorously, and did careful controls to determine whether BAF, a tiny (89-aa) protein, was disrupted by fusion to proteins such as GFP or TurboID. Animals carrying the G12T mutation exhibited reduced lifespan (Fig. 1, S1), lower responses to UV irradiation and heat-stress (Fig 7B, 7C), and revealed unexpected germline-specific defects in male worms (Fig. S2, S3), and altered gene expression in two tissues affected by human HGPS (Figs. 4 and 5). Overall, this manuscript strongly supports the major conclusion that this *C. elegans* line is a powerful model for human NGPS that complements a previously reported *Drosophila* model.

Equally importantly, from the viewpoint of fundamental discovery, this manuscript also reports major advances in understanding how BAF influences gene expression at the molecular level. Through careful attention to controls, and experimental design, the authors overcome many complications that make BAF difficult to study: its essential roles in mitosis and early embryogenesis, the 'tag'-sensitivity of endogenous BAF, and the absolute necessity to study BAF in native cell types. The authors carefully compared the impacts of tagging either baf-1 or lamin, and compared wildtype versus G12T-mutated baf-1 interactions with lamin and emerin (Fig. S5, S6, S8, S9; videos S1 and S2). DamID-Baf-1 access to chromatin was unaffected by the G12T mutation (Fig. S7), but they successfully identified subsets of genes 'occupied' by baf-1 in specific cell types, some of which were significantly affected in opposite ways by the NGPS mutation (Fig. 4, Fig. 5). However, these important new results are described too briefly, and discussion is inadequate. E.g., in hypodermal cells, the baf-1 G12T mutation dysregulated genes encoding proteins in five categories (ribosomal, proton transport, cuticle components, cell surface, lysine acetylation), by downregulating genes in three categories (ribosomal, proton transport, histone acetylation) and upregulating three other categories (cuticle components, cell surface, apical region).

In intestinal cells, the mutation dysregulated genes in 8 categories (ribosomal, response to X-ray, proton transport, proteasome binding, mitochondrial protein import, endopeptidase activity, carboxy-lyase activity, ATP generation), by downregulating genes in 5 categories (ribosomal, proton transport, peptidase activity, NAD binding, metal cluster binding) and upregulating 3 categories (ribosomal, response to external stimulation, histone acetylation). Opposite results for "ribosomal genes" is confusing. Examples of genes in each affected category are shown in Fig. 6. To fully interpret this data, and address apparently-conflicting results, further analysis is needed to determine if any affected groups of genes have shared regulators. For example, Fig 5E shows "ribosomal protein genes" are both up- and down-regulated by the mutation. The authors should consider: (a) WHICH ribosomal genes are in each category, and (b) does either group of genes have known regulators that might be differentially affected by the baf-1 mutation? Similar consideration of other sets of differentially-affected genes might provide novel insight into specific chromatin-regulatory proteins (e.g., potential baf-1 partners; see next paragraph) affected by the

NGPS mutation.

The current manuscript is too strictly focused on establishing *C. elegans* as a model for NGPS, and neglects the novel discoveries. The authors did not consider or discuss HOW a baf-1 mutation might cause such complex gene expression outcomes- given that baf-1 binds dsDNA nonspecifically. One plausible molecular explanation is that the NGPS mutation might affect baf-1 interactions with: (a) transcription factors (Requiem, RBBP4, DDB1) or chromatin-regulators (PARP1; UV-regulated interactions with DDB2 and CUL4A) identified as BAF-associated in a proteomic study (Montes de Oca et al., 2009), or (b) histone modifiers such as SET/I2PP2A (blocks H3 dephosphorylation) or H3K9 methyltransferase 'G9a' (Montes de Oca et al., 2011), or (c) other regulators that control affected genes identified in this manuscript.

Other clarifications and revisions to improve the manuscript:

Figures 1, 2, 4, 5: the graphs in Fig 1A,B,D-F and Fig 2B,D and the colorscales in Fig 4F and Fig 5E are uninterpretable when printed in black-and-white. Please fix Figs 1 and 2 using black/light-gray/white/stippled for bar graphs, and black/light-gray/solid/dotted/dashed for line graphs. Fig 2B can be fixed by direct-labeling of class numbers within each bar (instead of 'color-coding' separately).

Revise abstract lines 40-42 ("suggesting a direct relationship between BAF-1 binding [to what?] and gene expression") to reflect the deeper analysis.

Lines 132-155 (Figure 1): The impact on sperm production suggests the NGPS mutation might affect association with Germ cell-less (GCL), a transcription repressor that competes with BAF for binding to emerin in mammalian cells (Holaska et al., 2003 JBC).

Lines 151-154: Did not understand the fog-2 'feminized worm' experiments. Please briefly explain for non-worm experts.

Line 190: Clarify that nuclear shapes were categorized manually by single-blind observer.

Line 237-252: Abnormal chromosome segregation and postmitotic nuclear assembly in all *gfp::baf-1(G12T)* embryos is fully consistent (not 'presumably causative'; line 251) with the embryonic loss-of-function phenotype for baf-1 (Margalit et al., 2005, PNAS) and is consistent with mutational disruption of binding to lamin (Liu J et al., 2000, MBC) and/or LEM-domain proteins (Liu J, Lee KK et al., 2003, PNAS).

Line 424: Agree that this new *C. elegans* model is important and strongly complements the *Drosophila* NGPS model.

Lines 463-464: Agree that future suppressor analysis in this *C. elegans* model will be powerfully informative.

Lines 530-533: Baf-1 localization (mobility) in intestinal cells is known to change profoundly in response to heat shock, caloric restriction or food deprivation (Bar et al., 2014, MBC). It would be worthwhile testing, in future, whether the NGPS mutation affects baf-1 localization in response to these stresses.

Referees cross-commenting

I agree with the comments from both other reviewers.

2. Significance:

Significance (Required)

Overall, this manuscript strongly supports the major conclusion that this *C. elegans* line is a powerful model for human NGPS that complements a previously reported *Drosophila* model.

3. How much time do you estimate the authors will need to complete the suggested revisions:

Estimated time to Complete Revisions (Required)

(Decision Recommendation)

Cannot tell / Not applicable

Yes

Full Revision

Manuscript number: RC-2024-02430

Corresponding author(s): Peter, Askjaer

[Please use this template only if the submitted manuscript should be considered by the affiliate journal as a full revision in response to the points raised by the reviewers.]

*If you wish to submit a preliminary revision with a revision plan, please use our "Revision Plan" template. **It is important to use the appropriate template to clearly inform the editors of your intentions.**]*

1. General Statements [optional]

This section is optional. Insert here any general statements you wish to make about the goal of the study or about the reviews.

This section is mandatory. Please insert a point-by-point reply describing the revisions that were already carried out and included in the transferred manuscript.

We thank the reviewers for their positive and constructive criticism. We answer their points one by one below (in blue text).

Reviewer #1

1.) In the *baf-1* G12T mutants the authors find reduced levels of lamin in hypodermal nuclei. It would be good to also examine the dynamics of lamin in the second tissue that was subjected to DamID (intestinal cells).

We provide a complete analysis of GFP::LMN-1 and EMR-1::mCh in control and *baf-1(G12T)* day 1 adults in intestine and hypodermis and at 20°C and 25°C. These data demonstrate that GFP::LMN-1 expression is reduced in *baf-1(G12T)* mutants in both tissue and at both temperatures. In contrast, for EMR-1::mCh a significant reduction was only observed in hypodermal nuclei at 20°C.

The effects on GFP::LMN-1 and EMR-1::mCh in the hypodermis 20°C were reported in Figure 2E-F in the original version of our manuscript. We have moved these data to the new Supplementary Figure S5 and represent instead the data obtained for hypodermis at 25°C in Figure 2E-F for consistency with the data represented in Figure 2A-D. Data on intestine for both markers and both temperatures are also included in the new Supplementary Figure S5.

We have modified the text as follows (original text in blue; new text in red):

“To test the impact of *baf-1(G12T)* on LMN-1, EMR-1, and BAF-1 localization *in vivo*, we quantified these factors at the NE of hypodermal and intestinal cells. We observed a significantly lower median GFP::LMN-1 signal at the NE in *baf-1(G12T)* mutants in both tissues at 20°C and 25°C (Figure 2E; Supplementary Figure S5A-C). In contrast, accumulation of EMR-1 at the NE was unaffected by the *baf-1(G12T)* mutation in both tissues at 25°C and reduced in the hypodermis at 20°C (Figure 2F; Supplementary Figure S5D-F). In human NGPS cells, emerlin was observed to be delocalized to the ER (Janssen et al., 2022; Puente et al., 2011), but we detected no increase in cytoplasmic EMR-1::mCh signal in the mutant, indicating that this NGPS phenotype is not present in the *C. elegans* model. In agreement with these microscopy data, analysis of whole-worm mRNA levels by quantitative RT-PCR also revealed a significant reduction in *lmn-1* expression whereas *emr-1* was unaffected (Supplementary Figure S4E-F).”

2.) The authors make a statement that EMR-1 expression was reduced in the *baf-1 G12T* mutant, but do not comment on LMN-1 expression. Can a statement on this be made by RT PCR?

Our gene expression analysis by RAPID determined a significant reduction in *emr-1* expression in the intestine of *baf-1(G12T)* mutants, using a fold change of 2 as threshold. In contrast, expression of *emr-1* in hypodermis as well as *baf-1* and *lmn-1* expression in both tissues were not significantly different between wild type and *baf-1(G12T)* mutants in our RAPID data. We performed qRT-PCR on bulk mRNA to compare the expression of *baf-1*, *emr-1* and *lmn-1* in control versus *baf-1(G12T)* mutants. No differences were detected for *baf-1* and *emr-1* (new Supplementary Figure S4E-F). Considering that the qRT-PCR is on bulk mRNA, the *emr-1* result is compatible with the RAPID data that suggest deregulation of *emr-1* only in intestine and unaffected expression in the hypodermis. For *baf-1* there is agreement between qRT-PCR and RAPID data from both tissues (no difference in the mutant). For *lmn-1*, the qRT-PCR analysis suggests a modest reduction (23%; not reaching the threshold applied in the RAPID analysis) in *baf-1(G12T)* mutants, which is concordant with the reduction observed in GFP::LMN-1 intensity in hypodermis and intestine by confocal microscopy (e.g. 14% reduction in median GFP::LMN-1 intensity in hypodermis at 25°C; Figure 2E).

The discordance between RAPID and live imaging for *emr-1/EMR-1::mCh* (a reduction in the intestine or the hypodermis according to RAPID or live imaging, respectively) is not surprising. Although mRNA and protein levels in general correlate well, often, variation in transcription can only explain <50% of variation in protein level (Vogel & Marcotte, 2012). Processes like mRNA and protein turnover as well as translational regulation can “uncouple” transcriptomics from proteomics. Nevertheless, transcriptomics analyses (RNA-seq, DamID-based, etc.) are relevant to determine gene regulatory pathways, effects of mutations, etc.

We have added these two sentences to the manuscript:

“In agreement with these microscopy data, analysis of whole-worm mRNA levels by quantitative RT-PCR also revealed a significant reduction in *lmn-1* expression whereas *emr-1* was unaffected (**Supplementary Figure S4E-F**).”

“As described above, the amount of endogenously tagged EMR-1::mCh at the NE of intestinal cells was normal in *baf-1(G12T)* mutants (**Supplementary Figure S5F**), suggesting a cellular capacity to buffer the downregulation of *emr-1* transcription (Vogel & Marcotte, 2012).”

3.) The authors find few alterations in gene regulation of the loci which have different occupancy WT BAF-1 versus BAF-1 G12T. It was surprising to see the DamID and RNA polymerase DamID experiments be done with worms grown at 20°C, because the more penetrant phenotypes at the organismal level were observed at 25°C. Could this be the reason for the little change of chromatin occupancy of BAF-1 and BAF-1 G12T or few changes in gene expression? Would it make sense to examine the expression of some selected BAF-1 bound loci by single molecule Fish at 25°C and compare expression wt versus *baf-1* G12T?

We performed the DamID experiments at 20°C to avoid potential artifacts and/or toxicity by higher expression levels of Dam fusion proteins (Greil, Moorman, & van Steensel, 2006; Schuster et al., 2010). We note that altered UV and tert-butyl hydroperoxide was observed at 20°C, indicating that the *baf-1(G12T)* allele affects physiology at several temperatures. The original version of our manuscript described the expression of fluorescently tagged LMN-1 and EMR-1 in the hypodermis at 20°C (Figure 2E-F). As described above, in the revised version, we report the expression in the intestine at 20°C and in both tissues at 25°C. For GFP::LMN-1, a similar reduction in the *baf-1(G12T)* mutant was observed at the two temperatures in both tissues, whereas for EMR-1::mCh a reduction was only seen in the hypodermis at 20°C. Taken together, we conclude that 20°C is a suitable temperature for the DamID experiments.

We appreciate the suggestion to study expression of genes bound by BAF-1 by smFISH. However, we anticipate that because the hypodermis is composed mostly of large syncytia covering the round body of the animal, smFISH would be difficult to quantify. Regarding loci with different occupancy of WT BAF-1 versus BAF-1(G12T), the *emr-1* locus was bound in the intestine by Dam::BAF-1 but not by Dam::BAF-1(G12T) (Figure 6B). As mentioned above, we observed that *emr-1* expression was reduced in intestine of *baf-1(G12T)* mutants, suggesting that BAF-1 binding has a positive effect of transcription of this locus.

4.) The finding that BAF-1 nuclear envelope localization remains unchanged in the mutant stems from detection of the inserted GFP epitope. Given that the tag has an influence on the BAF-1 G12 12T mutant viability, this statement should be phrased with more care. The tag could influence the turnover of the protein for example. Maybe Western blots comparing the signal of WT BAF-1 worms and BAF-1 G12T mutant worms would be instructive to compare the levels of the protein (at 20oC and at 25oC, day 1 adults and day 8 adults).

We performed Western blot experiments to address this. As controls, we included strains expressing equal amounts of GFP::BAF-1 and GFP::BAF-1(G12T) strains (Figure 3E and Supplementary Figure 7 in original manuscript reported equal expression of the two proteins). Surprisingly, the polyclonal anti-BAF-1 serum raised against recombinant, full-length wild type BAF-1 (Gorjanacz et al., 2007) has significantly lower affinity for mutant GFP::BAF-1(G12T) than for GFP::BAF-1, which precludes the evaluation of untagged proteins:

Figure 1. Western blot analyses with anti-BAF-1 serum (Gorjanacz et al, 2007). (A) Embryonic extracts. A band of the expected size is observed in wildtype embryos (*), but not in *baf-1(G12T)* embryos. (B) Extracts from young adults. A faint band of the expected size is observed in wildtype embryos (* in lane 1; longer exposure is shown below), whereas a more prominent band is present corresponding to endogenously tagged GFP::BAF-1 (** in lane 2). The intensity of the

potential GFP::BAF-1(G12T) is reduced by >80% (lane 4; >90% reduction was observed in a second experiment).

We point out in the revised manuscript that the conclusion on equal BAF-1 and BAF-1(G12T) expression was based on endogenously tagged proteins: “Quantifying the intensity at the NE or in the nucleoplasm of hypodermal cells did not demonstrate any difference between endogenously GFP-tagged wild-type and mutant BAF-1 (Figure 3E). A small reduction in cytoplasmic signal was observed for BAF-1(G12T), however, no difference was detected in the ratio between nucleoplasmic/cytoplasmic signal (Figure 3E). Quantitative RT-PCR analysis of whole-worm RNA samples also indicated that *baf-1* and *baf-1(G12T)* are expressed at identical levels (Supplementary Figure S4E-F).”

5.) Line 105: typo: remove "s"

Corrected.

6.) Line 154: A conclusion is missing for the *fog-2* experiment.

We have modified the text as follows (original text in blue; new text in red): “To test this possibility, we incubated *baf-1(G12T)* males with *fog-2(q71)* feminized worms that only produce oocytes and counted daily offspring. At 25°C, the *fog-2(q71)* allele prevents spermatogenesis specifically in XX hermaphrodites whereas X0 males are unaffected (Schedl & Kimble, 1988).

Full Revision

We observed a reduction in brood size of approximately one third when sperm came from *baf-1(G12T)* males (**Supplementary Figure S2B, C**). Thus, we concluded that the *baf-1(G12T)* mutation has a negative impact on spermatogenesis. The male/female ratio in the progeny was ~1, suggesting that meiotic segregation of chromosomes was normal in *baf-1(G12T)* males.”

7). Would it make sense to discuss a possible influence of altered lamin binding to the nuclear envelope in the mutant in the context of the gene expression results?

We agree that this point is relevant, and we have added the following text to the Discussion: “At current, we can only speculate about how the NGPS mutation might affect gene expression. Proteomics analyses indicate that BAF interacts with several histones and transcription factors (Montes de Oca, Shoemaker, Gucek, Cole, & Wilson, 2009), and the differences between BAF-1 and BAF-1(G12T)’s chromatin binding profiles reported here might be accompanied by changes in the association of chromatin factors at the deregulated loci. A particularly interesting candidate is GCL-1/germ cell-less 1, a repressive factor involved in spermatogenesis (Holaska, Lee, Kowalski, & Wilson, 2003). Moreover, it is plausible that the diminished recruitment of LMN-1 to the NE in *baf-1(G12T)* mutants modifies its interaction with the genome and with chromatin factors.”

8). In a nutshell, the authors have established a convincing accessible model system for studying aging, ready for consecutive testing interventions to reduce the pace of premature aging.

We appreciate and share the opinion of the reviewer.

Reviewer #2

1). The value of this work is two-fold: First, it is a very robust characterization of NGPS worms. Second, this will be a very useful model for the study of NGPS. Overall, the study is well-designed, technically strong, and the results are carefully and thoughtfully interpreted, which is nicely exemplified by the discussion of the relatively small number of genes which are differentially bound by BAF1 and are also differentially expressed and the authors do a good job of not overinterpreting the data, but simply state them. The results are convincing and informative.

We thank the reviewer for her/his positive evaluation.

My only minor point that may make this paper marginally better is that it would be nice to have a paragraph in the Discussion elaborating on the potential and the limitations of using the worm model to understand human NGPS, for example, humans have multiple lamin proteins etc.

We agree with the reviewer and have added the following text to the Discussion: “We note that the simplicity of invertebrates also implies certain limitations. For instance, while both human and *C. elegans* genomes contain a single BAF gene, humans, but not *C. elegans*, express

multiple lamin isoforms in tissue-specific ratios that regulate chromatin organization and nuclear mechanics (Swift et al., 2013). Thus, *C. elegans* is not suitable to explore potential differences in how wild type and NGPS BAF interacts differently with the various lamin isoforms.”

Reviewer #3

1). Overall, this manuscript strongly supports the major conclusion that this *C. elegans* line is a powerful model for human NGPS that complements a previously reported *Drosophila* model. Equally importantly, from the viewpoint of fundamental discovery, this manuscript also reports major advances in understanding how BAF influences gene expression at the molecular level.

We thank the reviewer for her/his positive evaluation.

2). DamID-Baf-1 access to chromatin was unaffected by the G12T mutation (Fig. S7), but they successfully identified subsets of genes 'occupied' by baf-1 in specific cell types, some of which were significantly affected in opposite ways by the NGPS mutation (Fig. 4, Fig. 5). However, these important new results are described too briefly, and discussion is inadequate. E.g., in hypodermal cells, the baf-1 G12T mutation dysregulated genes encoding proteins in five categories (ribosomal, proton transport, cuticle components, cell surface, lysine acetylation), by downregulating genes in three categories (ribosomal, proton transport, histone acetylation) and upregulating three other categories (cuticle components, cell surface, apical region). In intestinal cells, the mutation dysregulated genes in 8 categories (ribosomal, response to X-ray, proton transport, proteasome binding, mitochondrial protein import, endopeptidase activity, carboxy-lyase activity, ATP generation), by downregulating genes in 5 categories (ribosomal, proton transport, peptidase activity, NAD binding, metal cluster binding) and upregulating 3 categories (ribosomal, response to external stimulation, histone acetylation). Opposite results for "ribosomal genes" is confusing. Examples of genes in each affected category are shown in Fig. 6. To fully interpret this data, and address apparently-conflicting results, further analysis is needed to determine if any affected groups of genes have shared regulators. For example, Fig 5E shows "ribosomal protein genes" are both up- and down-regulated by the mutation. The authors should consider: (a) WHICH ribosomal genes are in each category, and (b) does either group of genes have known regulators that might be differentially affected by the baf-1 mutation? Similar consideration of other sets of differentially-affected genes might provide novel insight into specific chromatin-regulatory proteins (e.g., potential baf-1 partners; see next paragraph) affected by the NGPS mutation.

At first it may seem confusing that some ribosomal genes are downregulated while others are upregulated. However, the *baf-1(G12T)* mutant represents a disease situation and not a process of natural selection where one might expect "meaningful" groups of up- and down-regulated genes. We have looked closer at the individual deregulated ribosomal genes and found genes encoding structural components of large ribosomal subunits that are either upregulated (*rpl-10*, *rpl-29*, *rpl-36*) or downregulated (*rpl-1*, *rpl-3*, *rpl-30*) in the intestine. Although these opposite behaviors might seem confusing, we propose that they reflect deregulation of ribosome biosynthesis, which is in concordance with the observations in NGPS

fibroblasts (Breusegem et al., 2022). We agree that it will be important to investigate how the NGPS mutation induces these oppositely directed effects on gene expression. We found a significant higher association of the 13 deregulated ribosomal genes to BAF-1(G12) than to BAF-1 in the intestine, but we believe it goes beyond the scope of this manuscript to focus on the underlying mechanisms.

3). The current manuscript is too strictly focused on establishing *C. elegans* as a model for NGPS, and neglects the novel discoveries. The authors did not consider or discuss HOW a baf-1 mutation might cause such complex gene expression outcomes- given that baf-1 binds dsDNA nonspecifically. One plausible molecular explanation is that the NGPS mutation might affect baf-1 interactions with: (a) transcription factors (Requiem, RBBP4, DDB1) or chromatin-regulators (PARP1; UV-regulated interactions with DDB2 and CUL4A) identified as BAF-associated in a proteomic study (Montes de Oca et al., 2009), or (b) histone modifiers such as SET/I2PP2A (blocks H3 dephosphorylation) or H3K9 methyltransferase 'G9a' (Montes de Oca et al., 2011), or (c) other regulators that control affected genes identified in this manuscript.

We agree that this point is very relevant, but at this point we do not have experimental support for any of these possibilities. As indicated in the response to Reviewer #2, we have added the following text to the Discussion: “At current, we can only speculate about how the NGPS mutation might affect gene expression. Proteomics analyses indicate that BAF interacts with several histones and transcription factors (Montes de Oca et al., 2009), and the differences between BAF-1 and BAF-1(G12T)'s chromatin binding profiles reported here might be accompanied by changes in the association of chromatin factors at the deregulated loci. A particularly interesting candidate is GCL-1/germ cell-less 1, a repressive factor involved in spermatogenesis (Holaska et al., 2003). Moreover, it is plausible that the diminished recruitment of LMN-1 to the NE in baf-1(G12T) mutants modifies its interaction with the genome and with chromatin factors.”

4). Figures 1, 2, 4, 5: the graphs in Fig 1A,B,D-F and Fig 2B,D and the colorscales in Fig 4F and Fig 5E are uninterpretable when printed in black-and-white. Please fix Figs 1 and 2 using black/light-gray/white/stippled for bar graphs, and black/light-gray/solid/dotted/dashed for line graphs. Fig 2B can be fixed by direct-labeling of class numbers within each bar (instead of 'color-coding' separately).

We thank the reviewer for this suggestion. We have modified the figures to enable better visualization when printed in BW.

5). Revise abstract lines 40-42 ("suggesting a direct relationship between BAF-1 binding [to what?] and gene expression") to reflect the deeper analysis.

We have rephrased this sentence, so it now reads: “Most genes deregulated by the *baf-1(G12T)* mutation were characterized by a change in BAF-1 association, suggesting a direct relation

Full Revision

between association of a gene to BAF-1 and its expression.” However, we prefer to not extend into speculations in the abstract because of lack of experimental evidence.

6). Lines 132-155 (Figure 1): The impact on sperm production suggests the NGPS mutation might affect association with Germ cell-less (GCL), a transcription repressor that competes with BAF for binding to emerlin in mammalian cells (Holaska et al., 2003 JBC).

This is indeed an interesting possibility and we have incorporated it into to Discussion (see answer to point 3 above).

7). Lines 151-154: Did not understand the fog-2 'feminized worm' experiments. Please briefly explain for non-worm experts.

Please see our response to Reviewer #1's point 6.

8). Line 190: Clarify that nuclear shapes were categorized manually by single-blind observer.

We have amended the text: “Nuclei were manually classified by single-blind observer based on their morphology as previously described (Perez-Jimenez, Rodriguez-Palero, Rodenas, Askjaer, & Munoz, 2014), except that we introduced a fourth class to describe the most irregular nuclei (see Materials and Methods).”

9). Line 237-252: Abnormal chromosome segregation and postmitotic nuclear assembly in all *gfp::baf-1(G12T)* embryos is fully consistent (not 'presumably causative'; line 251) with the embryonic loss-of-function phenotype for *baf-1* (Margalit et al., 2005, PNAS) and is consistent with mutational disruption of binding to lamin (Liu J et al., 2000, MBC) and/or LEM-domain proteins (Liu J, Lee KK et al., 2003, PNAS).

We thank the reviewer for pointing this out. We have added the following sentence: “These phenotypes are consistent with the effects of embryonic depletion of BAF-1 or LMN-1 (Liu et al., 2000; Margalit, Segura-Totten, Gruenbaum, & Wilson, 2005).”

10). Lines 530-533: Baf-1 localization (mobility) in intestinal cells is known to change profoundly in response to heat shock, caloric restriction or food deprivation (Bar et al., 2014, MBC). It would be worthwhile testing, in future, whether the NGPS mutation affects *baf-1* localization in response to these stresses.

We appreciate this suggestion, and we agree with the reviewer that it would be important to test this in future studies.

Other changes:

Missing column in Table S3 added.

Mistake if column heading in Table S4 corrected.

- Breusegem, S. Y., Houghton, J., Romero-Bueno, R., Fragoso-Luna, A., Kentistou, K. A., Ong, K. K., . . . Larrieu, D. (2022). A multiparametric anti-aging CRISPR screen uncovers a role for BAF in protein translation. *bioRxiv*. doi:10.1101/2022.10.07.509469
- Gorjanacz, M., Klerkx, E. P., Galy, V., Santarella, R., Lopez-Iglesias, C., Askjaer, P., & Mattaj, I. W. (2007). Caenorhabditis elegans BAF-1 and its kinase VRK-1 participate directly in post-mitotic nuclear envelope assembly. *Embo J*, 26(1), 132-143. doi:10.1038/sj.emboj.7601470
- Greil, F., Moorman, C., & van Steensel, B. (2006). DamID: mapping of in vivo protein-genome interactions using tethered DNA adenine methyltransferase. *Methods Enzymol*, 410, 342-359. doi:10.1016/S0076-6879(06)10016-6
- Holaska, J. M., Lee, K. K., Kowalski, A. K., & Wilson, K. L. (2003). Transcriptional repressor germ cell-less (GCL) and barrier to autointegration factor (BAF) compete for binding to emerin in vitro. *J Biol Chem*, 278(9), 6969-6975.
- Janssen, A., Marcelot, A., Breusegem, S., Legrand, P., Zinn-Justin, S., & Larrieu, D. (2022). The BAF A12T mutation disrupts lamin A/C interaction, impairing robust repair of nuclear envelope ruptures in Nestor-Guillermo progeria syndrome cells. *Nucleic Acids Res*. doi:10.1093/nar/gkac726
- Liu, J., Rolef Ben-Shahar, T., Riemer, D., Treinin, M., Spann, P., Weber, K., . . . Gruenbaum, Y. (2000). Essential roles for Caenorhabditis elegans lamin gene in nuclear organization, cell cycle progression, and spatial organization of nuclear pore complexes. *Mol Biol Cell*, 11(11), 3937-3947.
- Margalit, A., Segura-Totten, M., Gruenbaum, Y., & Wilson, K. L. (2005). Barrier-to-autointegration factor is required to segregate and enclose chromosomes within the nuclear envelope and assemble the nuclear lamina. *Proc Natl Acad Sci U S A*, 102(9), 3290-3295. doi:10.1073/pnas.0408364102
- Montes de Oca, R., Shoemaker, C. J., Gucek, M., Cole, R. N., & Wilson, K. L. (2009). Barrier-to-autointegration factor proteome reveals chromatin-regulatory partners. *PLoS ONE*, 4(9), e7050. doi:10.1371/journal.pone.0007050
- Perez-Jimenez, M. M., Rodriguez-Palero, M. J., Rodenas, E., Askjaer, P., & Munoz, M. J. (2014). Age-dependent changes of nuclear morphology are uncoupled from longevity in Caenorhabditis elegans IGF/insulin receptor daf-2 mutants. *Biogerontology*, 15(3), 279-288. doi:10.1007/s10522-014-9497-0
- Puente, X. S., Quesada, V., Osorio, F. G., Cabanillas, R., Cadinanos, J., Fraile, J. M., . . . Lopez-Otin, C. (2011). Exome sequencing and functional analysis identifies BANF1 mutation as the cause of a hereditary progeroid syndrome. *Am J Hum Genet*, 88(5), 650-656. doi:10.1016/j.ajhg.2011.04.010
- Schedl, T., & Kimble, J. (1988). fog-2, a germ-line-specific sex determination gene required for hermaphrodite spermatogenesis in Caenorhabditis elegans. *Genetics*, 119(1), 43-61. doi:10.1093/genetics/119.1.43
- Schuster, E., McElwee, J. J., Tullet, J. M., Doonan, R., Matthijssens, F., Reece-Hoyes, J. S., . . . Gems, D. (2010). DamID in C. elegans reveals longevity-associated targets of DAF-16/FoxO. *Mol Syst Biol*, 6, 399. doi:10.1038/msb.2010.54
- Swift, J., Ivanovska, I. L., Buxboim, A., Harada, T., Dingal, P. C., Pinter, J., . . . Discher, D. E. (2013). Nuclear lamin-A scales with tissue stiffness and enhances matrix-directed differentiation. *Science*, 341(6149), 1240104. doi:10.1126/science.1240104

Full Revision

Vogel, C., & Marcotte, E. M. (2012). Insights into the regulation of protein abundance from proteomic and transcriptomic analyses. *Nat Rev Genet*, 13(4), 227-232.
doi:10.1038/nrg3185

Dear Peter,

Congratulations on a great revision! Overall, the referees have been positive. However, there remain a few editorial items to attend to in a revised version. When you submit your revision, please add these changes to a new point-by-point response:

1. Please provide a completed author checklist.
2. Please upload the manuscript in .docx format, main figures should be removed from the manuscript and upload as individual, high resolution figure files.
3. Please reduce the number of keywords to 5.
4. Please add a Data Availability Section as described in our author guidelines.
5. Please merge the funding with the acknowledgements section.
6. Please consider adding the Karolinska Institutet Doctoral Education Grant to the list of funders in our online system
7. Please review our new policy on conflict of interests on the EMBO author guide website and update the title of this section to: Disclosure and competing interests statement.
8. Suppl. videos should be renamed "Movie EV1" and "Movie EV2". Their legends should be removed from the manuscript text and zipped to the corresponding movie file.
9. Please provide a reagents and tools table.
10. We require the publication of source data, particularly for electrophoretic gels and blots and graphs, with the aim of making primary data more accessible and transparent to the reader. It would be great if you could provide me with a PDF file per figure that contains the original, uncropped and unprocessed scans of all or key gels used in the figure or for graphs, an Excel spreadsheet with the original data used to generate the graphs. The PDF files should be labeled with the appropriate figure/panel number, and should have molecular weight marker; further annotation could be useful but is not essential. The PDF files will be published online with the article as supplementary "Source Data" files.
11. We do not allow unpublished data to be referenced in the manuscript, please remove these "data not shown" from page 10 and 11.
12. We include a synopsis of the paper (see <http://emboj.embopress.org/>). Please provide me with a general summary statement and 3-5 bullet points that capture the key findings of the paper.
13. We also need a summary figure for the synopsis. The size should be 550 wide by 200-440 high (pixels). You can also use something from the figures if that is easier.
14. Up to five suppl. figures can be renamed "Figure EV1" - 5 and uploaded as individual, high res figure files. Legends should be in the manuscript, after the main figure legends, and under the heading "Expanded View Figure Legends". The remaining suppl. figures should be compiled with legends, in a PDF labelled "Appendix", with the nomenclature "Appendix Figure S1" etc. The appendix will need a table of contents, including page numbers.
15. Suppl. tables S1,2,5-7 should be renamed "Table EV1" etc. Suppl.tables S3 and 4 should be renamed "Dataset EV1" and 2, and their legends should be added to the excel files in a separate tab.
16. Please note that the legends for figures 3b-c is not provided in the sequential manner (legend for figure 3c is provided before legend of figure 3b). This needs to be rectified.
17. Please note that the exact p values are not provided in the legends of figures 1b, d, e-f; 3e; 4c; 5a-b; 7a.
18. Please indicate the statistical test used for data analysis in the legends of figures 4f; 5e.
19. Please note that in figures 1b, d, e-f; 2e; there is a mismatch between the annotated p values in the figure legend and the annotated p values in the figure file that should be corrected.
20. Please note that for the figure 1a, p-values and statistical tests are indicated in the legends. However, comparison for the same, "****/****/**/*" has not been represented in the figure. Please rectify this in the figure or legend as applicable.

21. Please note that the box plots need to be defined in terms of minima, maxima, centre, bounds of box and whiskers, and percentile in the legends of figures 3e; 4c; 5a-b; 6a-b.

22. Please note that information related to n is missing in the legends of figures 4c; 5c-d; 6a-c.

23. Please note that the error bars are not defined in the legends of figures 3a, c.

24. Please note that in figures 1c; 2a, c; 3b, d; the scale bar unit should be corrected from μM to μm (in the figure legend).

Thank you for the opportunity to consider your work for publication, I look forward to your revision.

Warm wishes,
Kelly

Kelly M Anderson, PhD
Editor, The EMBO Journal
k.anderson@embojournal.org

Please click on the link below to submit the revision online:

Link Not Available

Referee #1:

The authors have fully addressed my minor comment. Given the scope and nature of the reported results and the lack of mechanistic insight, I consider this manuscript a good fit for EMBO Reports, but less so for EMBO J.

Referee #2:

The authors have submitted a carefully revised manuscript and addressed all my questions.

Referee #3:

This revised manuscript satisfies all previous concerns. I fully support publication of this important work. And thank the authors for carefully addressing my concerns.

Rev_Com_number: RC-2024-02430

New_manu_number: EMBOJ-2024-118450-T

Corr_author: Askjaer

Title: A progeria-associated BAF-1 mutation modulates gene expression and accelerates aging in *C. elegans*

Point-by-point response

>>> 1. Please provide a completed author checklist.

Uploaded.

>>> 2. Please upload the manuscript in .docx format, main figures should be removed from the manuscript and upload as individual, high resolution figure files.

Uploaded.

>>> 3. Please reduce the number of keywords to 5.

Corrected.

>>> 4. Please add a Data Availability Section as described in our author guidelines.

Added.

>>> 5. Please merge the funding with the acknowledgements section.

Amended.

>>> 6. Please consider adding the Karolinska Institutet Doctoral Education Grant to the list of funders in our online system

Will be attempted during online submission.

>>> 7. Please review our new policy on conflict of interests on the EMBO author guide website and update the title of this section to: Disclosure and competing interests statement.

Added.

>>> 8. Suppl. videos should be renamed "Movie EV1" and "Movie EV2". Their legends should be removed from the manuscript text and zipped to the corresponding movie file.

Corrected.

>>> 9. Please provide a reagents and tools table.

Uploaded.

>>> 10. We require the publication of source data, particularly for electrophoretic gels and blots and graphs, with the aim of making primary data more accessible and transparent to the reader. It would be great if you could provide me with a PDF file per figure that contains the original, uncropped and unprocessed scans of all or key gels used in the figure or for graphs, an Excel spreadsheet with the original data used to generate the graphs. The PDF files should be labeled with the appropriate figure/panel number, and should have molecular weight marker; further annotation could be useful but is not essential. The PDF files will be published online with the article as supplementary "Source Data" files.

We have uploaded all requested Source Data.

>>> 11. We do not allow unpublished data to be referenced in the manuscript, please remove these "data not shown" from page 10 and 11.

Corrected.

>>> 12. We include a synopsis of the paper (see <http://emboj.embopress.org/>). Please provide me with a general summary statement and 3-5 bullet points that capture the key findings of the paper.

Uploaded.

>>> 13. We also need a summary figure for the synopsis. The size should be 550 wide by 200-440 high (pixels). You can also use something from the figures if that is easier.

Uploaded.

>>> 14. Up to five suppl. figures can be renamed "Figure EV1" - 5 and uploaded as individual, high res figure files. Legends should be in the manuscript, after the main figure legends, and under the heading "Expanded View Figure Legends". The remaining suppl. figures should be compiled with legends, in a PDF labelled "Appendix", with the nomenclature "Appendix Figure S1" etc. The appendix will need a table of contents, including page numbers.

Corrected.

>>> 15. Suppl. tables S1,2,5-7 should be renamed "Table EV1" etc. Suppl.tables S3 and 4 should be renamed "Dataset EV1" and 2, and their legends should be added to the excel files in a separate tab.

Corrected.

>>> 16. Please note that the legends for figures 3b-c is not provided in the sequential manner (legend for figure 3c is provided before legend of figure 3b). This needs to be rectified.

Corrected.

>>> 17. Please note that the exact p values are not provided in the legends of figures 1b, d, e-f; 3e; 4c; 5a-b; 7a.

We indicate the exact p values in the revised figures, except for p values ≥ 0.05 that we consider not significant (indicated as n.s. in figures and legends). R Studio does not provide exact p values $< 2.2e-16$ (or $< 2e-16$ depending on the R package); we indicate these values as $p < 2.2e-16$ (or $< 2e-16$).

>>> 18. Please indicate the statistical test used for data analysis in the legends of figures 4f; 5e.

Added.

>>> 19. Please note that in figures 1b, d, e-f; 2e; there is a mismatch between the annotated p values in the figure legend and the annotated p values in the figure file that should be corrected.

Corrected.

>>> 20. Please note that for the figure 1a, p-values and statistical tests are indicated in the legends. However, comparison for the same, "****/**/**/*" has not been represented in the figure. Please rectify this in the figure or legend as applicable.

Corrected.

>>> 21. Please note that the box plots need to be defined in terms of minima, maxima, centre, bounds of box and whiskers, and percentile in the legends of figures 3e; 4c; 5a-b; 6a-b.

Corrected.

>>> 22. Please note that information related to n is missing in the legends of figures 4c; 5c-d; 6a-c.

Corrected.

>>> 23. Please note that the error bars are not defined in the legends of figures 3a, c.

Corrected.

>>> 24. Please note that in figures 1c; 2a, c; 3b, d; the scale bar unit should be corrected from μM to μm (in the figure legend).

Corrected.

Dear Dr. Askjaer,

Congratulations on an excellent manuscript, I am pleased to inform you that your manuscript has been accepted for publication in the EMBO Journal. Thank you for your comprehensive response to the referee concerns and for providing detailed source data. It has been a pleasure to work with you to get this to the acceptance stage.

I will begin the final checks on your manuscript before submitting to the publisher next week. Once at the publisher, it will take about 3 weeks for your manuscript to be published online. As a reminder, the entire review process including referee concerns, and your point-by-point response will be available to readers.

I will be in touch throughout the final editorial process until publication. In the meantime, I hope you find time to celebrate!

Yours sincerely,
Kelly

Kelly M Anderson, PhD
Editor, The EMBO Journal
k.anderson@embojournal.org

Rev_Com_number: RC-2024-02430

New_manu_number: EMBOJ-2024-118450R

Corr_author: Askjaer

Title: A progeria-associated BAF-1 mutation modulates gene expression and accelerates aging in *C. elegans*